# CEP128 is involved in spermatogenesis in humans and mice

Xueguang Zhang[1,15], Lingbo Wang[2,15], Yongyi Ma[3,15], Yan Wang[1,4,15], Hongqian Liu[1,5,15], Mohan Liu[1,15], Lang Qin[1,4,15], Jinghong Li[1,4], Chuan Jiang[1], Xiaojian Zhang[6], Xudong Shan[7,8], Yuliang Liu[9], Jinsong Li [10], Yaqian Li[1], Rui Zheng[1], Yongkang Sun[1], Jianfeng Sun[11], Xiangyou Leng[1], Yan Liang[12], Feng Zhang [2,13✉], Xiaohui Jiang [1,14✉], Yihong Yang [1,4✉] & Ying Shen [1✉]

Centrosomal proteins are necessary components of the centrosome, a conserved eukaryotic organelle essential to the reproductive process. However, few centrosomal proteins have been genetically linked to fertility. Herein we identify a homozygous missense variant of *CEP128* (c.665 G > A [p.R222Q]) in two infertile males. Remarkably, male homozygous knock-in mice harboring the orthologous *CEP128*[R222Q] variant show anomalies in sperm morphology, count, and motility. Moreover, *Cep128* knock-out mice manifest male infertility associated with disrupted sperm quality. We observe defective sperm flagella in both homozygous *Cep128* KO and KI mice; the cilia development in other organs is normal— suggesting that *CEP128* variants predominantly affected the ciliogenesis in the testes. Mechanistically, CEP128 is involved in male reproduction via regulating the expression of genes and/or the phosphorylation of TGF-β/BMP-signalling members during spermatogenesis. Altogether, our findings unveil a crucial role for CEP128 in male fertility and provide important insights into the functions of centrosomal proteins in reproductive biology.

[1] Department of Obstetrics/Gynecology, Key Laboratory of Obstetric, Gynecologic and Pediatric Diseases and Birth Defects of Ministry of Education, West China Second University Hospital, Sichuan University, Chengdu 610041, China. [2] Shanghai Key Laboratory of Female Reproductive Endocrine Related Diseases, Institute of Reproduction and Development, Obstetrics and Gynecology Hospital of Fudan University, Shanghai 200011, China. [3] Department of Gynecology and Obstetrics, Southwest Hospital, Third Military Medical University (Army Medical University), Chongqing 400000, China. [4] Reproduction Medical Centre, West China Second University Hospital, Sichuan University, Chengdu 610041, China. [5] Medical Genetics Department/Prenatal Diagnostic Centre, West China Second University Hospital, Sichuan University, Chengdu 610041, China. [6] Sichuan Academy of Medical Sciences of Sichuan Provincial People's Hospital, Chengdu 610041, China. [7] West China School of Basic Medical Sciences and Forensic Medicine, Chengdu 610041, China. [8] School of Medical and Life Sciences, Chengdu University of Traditional Chinese Medicine, Chengdu 610041, China. [9] Chengdu Research Base of Giant Panda Breeding, Chengdu 610041, China. [10] CAS Center for Excellence in Molecular Cell Science, Chinese Academy of Sciences, Shanghai 200031, China. [11] Teaching Hospital of Chengdu University of TCM, Chengdu 610041, China. [12] Research Core Facility of West China Hospital, Sichuan University, Chengdu 610041, China. [13] Shanghai Key Laboratory of Female Reproductive Endocrine Related Diseases, Institute of Reproduction and Development, Fudan University, Shanghai 200011, China. [14] Human Sperm Bank, West China Second University Hospital, Sichuan University, Chengdu 610041, China. [15] These authors contributed equally: Xueguang Zhang, Lingbo Wang, Yongyi Ma, Yan Wang, Hongqian Liu, Mohan Liu, Lang Qin. ✉email: zhangfeng@fudan.edu.cn; jxh424@126.com; yyhpumc@foxmail.com; yingcaishen01@163.com

Spermatogenesis is a highly complex, multi-step process that takes place within the testicular seminiferous epithelium, and that can be broadly divided into the following stages: the mitotic division of spermatogonia, meiotic division of spermatocytes, and maturation of haploid spermatozoa. Factors that perturb this process lead to male infertility that is generally manifested as decreased sperm count (azoospermia or oligozoospermia), impaired sperm motility (asthenozoospermia), or a high proportion of morphologically abnormal sperm (teratozoospermia). In ~15% of infertile men, a genetic defect is the most likely underlying cause of the pathology[1,2]. Although numerous male infertility genes have been identified, most genetic causes of male sterility are currently uncharacterized[3].

The normal spermatozoon is a streamlined, motile cell that contains a head, a neck and a tail. In the neck region, there is a head-tail coupling apparatus, a centrosome-based structure consisting of two centrioles and associated components including segmented column (Sc), capitulum, and basal plate (Bp). As an important structure of spermatozoa, the centrosome is necessary for the formation of the sperm flagellum and the connection between the head and tail of the sperm, and also forms the major microtubule-organizing center of the zygote[4]. Indeed, male infertility caused by sperm morphological abnormalities such as head-neck defects, multiple morphological abnormalities of the flagella (MMAF), dysplasia of the fibrous sheath and globozoospermia is in many accompanied by centrosome dysfunction[5]. It has long been accepted that centrosome integrity is critically important for successful fertilization, but direct experimental evidence of this is limited[5].

The centrosomal proteins comprise the essential components of the centrosome, and the relationships between centrosomal proteins and diseases are being uncovered. With the observation of centrosome defects in a broad set of diseases, enormous amounts of data in the literature have proposed that most centrosomal proteins are associated with cancer, and several are related to microcephaly, lissencephaly, schizophrenia, dwarfism, spinocerebellar ataxia, polycystic kidney disease, obesity, diabetes, and so on[6–8]. However, one area in need of more research is the contribution of centrosomal proteins in male reproduction. To date, only a few studies have indicated that dysfunction of centrosomal proteins may result in male sterility. Loss of function variants of CEP135[9] and CEP112[10] have been suggested to cause MMAF and acephalic spermatozoa phenotypes respectively in humans, and Cep55[11] and Cep131[12] are related to mouse infertility. Therefore, researches on the link between centrosomal proteins and impaired fecundity are particularly significant.

In this study, a homozygous missense variant of CEP128 was identified in two siblings with primary male infertility related to cryptozoospermia from a consanguineous family through whole-exome sequencing. CEP128 encodes the centrosomal protein CEP128, which is a basic subdistal appendage protein and functions on the mother centriole for the organization of the centriolar microtubules[13,14]. We further generated a mouse model harboring the orthologous missense variant of CEP128 in human cases via CRISPR-Cas9 gene editing, and the homozygous Cep128 knock-in (KI) mice exhibited reduced sperm counts and spermatozoa with morphological abnormalities. Intriguingly, the male Cep128 knock-out (KO) mice were infertile due to impaired spermatogenesis. These data show that CEP128 plays a crucial role in male fertility. Mechanistically, proteomics analysis on the testes of Cep128 KO and KI mice suggested that CEP128 is a functional protein with an essential role in the reproductive process by regulating expressions of Wbp2nl, Rcbtb2, Prss55, Crisp1, Defb22, Sun5, Tssk4, phosphorylated-Rbl1 (p-Rbl1), p-Gata4, and p-Trim33, which are associated with sperm accessory structure and fertilization process.

## Results

### Identification of a homozygous missense CEP128 variant in two affected siblings with cryptozoospermia.
A family medical history disclosed two siblings in a consanguineous family who had suffered from primary male infertility for years. Initially, no spermatozoa were observed in the replicate wet preparations. After centrifugation at $3000 \times g$ for 15 min, only a few defective spermatozoa were found in the proband and his sibling, which had irregular flagella and aberrant heads (Fig. 1a and Supplementary Fig. 1a). Similar clear features were also observed in the sperm cells from the two siblings by scanning electron microscopy (SEM) (Fig. 1b). In addition, transmission electron microscopy (TEM) was employed to analyze the ultrastructure of spermatozoa from the two siblings. Strikingly, incomplete proximal centriole (PC) and distal centriole (DC), and a defective Sc were visible in the sperm-connecting piece in the affected individuals, although the Bp was present (Fig. 1c). Moreover, when compared to the integrated and regular "9 + 2" axonemal arrangement of the flagella of the sperm from the normal control, the spermatozoa from the two patients represented the lacking or disorganized central-pair microtubules (CPs) or the absent and disordered microtubules as well as the outer dense fibers (ODFs) in the flagellar midpiece; most of the axonemal microtubules were missing and irregularly arranged in the principal piece; and the CPs or the peripheral microtubule doublets (MTDs) in the end piece were lacking (Fig. 1c and Supplementary Fig. 1b).

Whole-exome sequencing was next applied to this family (Fig. 2a). In total, 179 and 174 rare clinically relevant variants were detected in the proband and his sibling respectively, while none of them have been associated with male infertility (Supplementary Dataset 1). Thirteen rare functionally relevant homozygous variants were found in the proband and his sibling respectively (Supplementary Dataset 1). Among them, five homozygous variants are shared by the proband and the affected sibling. We then analyzed the possible functions of the genes affected by these brothers-shared homozygous variants. Importantly, a homozygous missense variant in CEP128 (c.665G > A [p.R222Q]) caught our attention. This variant is absent in most human populations according to the ExAC Browser, 1000 Genomes Project, and gnomAD databases (Supplementary Table 1). Low allele frequencies of this CEP128 variant were found in some East Asian populations (Japanese and Korean), while no homozygotes have been reported in human populations. However, neither heterozygous or homozygous variants of CEP128 (c.665G > A) were detected in our 1000 normal Chinese control males. Moreover, this variant is predicted to be deleterious by SIFT, PolyPhen-2, and CADD tools, and to be a polymorphism by MutationTaster (Supplementary Table 1). Notably, the affected site of the variant is highly conserved among different species (Fig. 2b). Furthermore, Sanger sequencing confirmed homozygosity for the CEP128 variant in the patients and the parents are heterozygous for the variant (Fig. 2c). To evaluate the influence of the identified p.R222Q variant on CEP128 expression, we performed immunofluorescence staining of semen smears from the patients and the normal control using the anti-CEP128 antibody. CEP128 colocalized with both PC and DC in the sperm of the control, while CEP128 signals were markedly enhanced in the sperm neck of the patients with the deficiency of tubulin staining labeling the centriole (Fig. 2d). Collectively, these findings suggest that the homozygous missense variant p.R222Q in CEP128 might contribute to the sperm defects in the two infertile patients.

### The effect of CEP128^R222Q variant on primary cilia development in cultured cells.
We further constructed Flag-wild-type

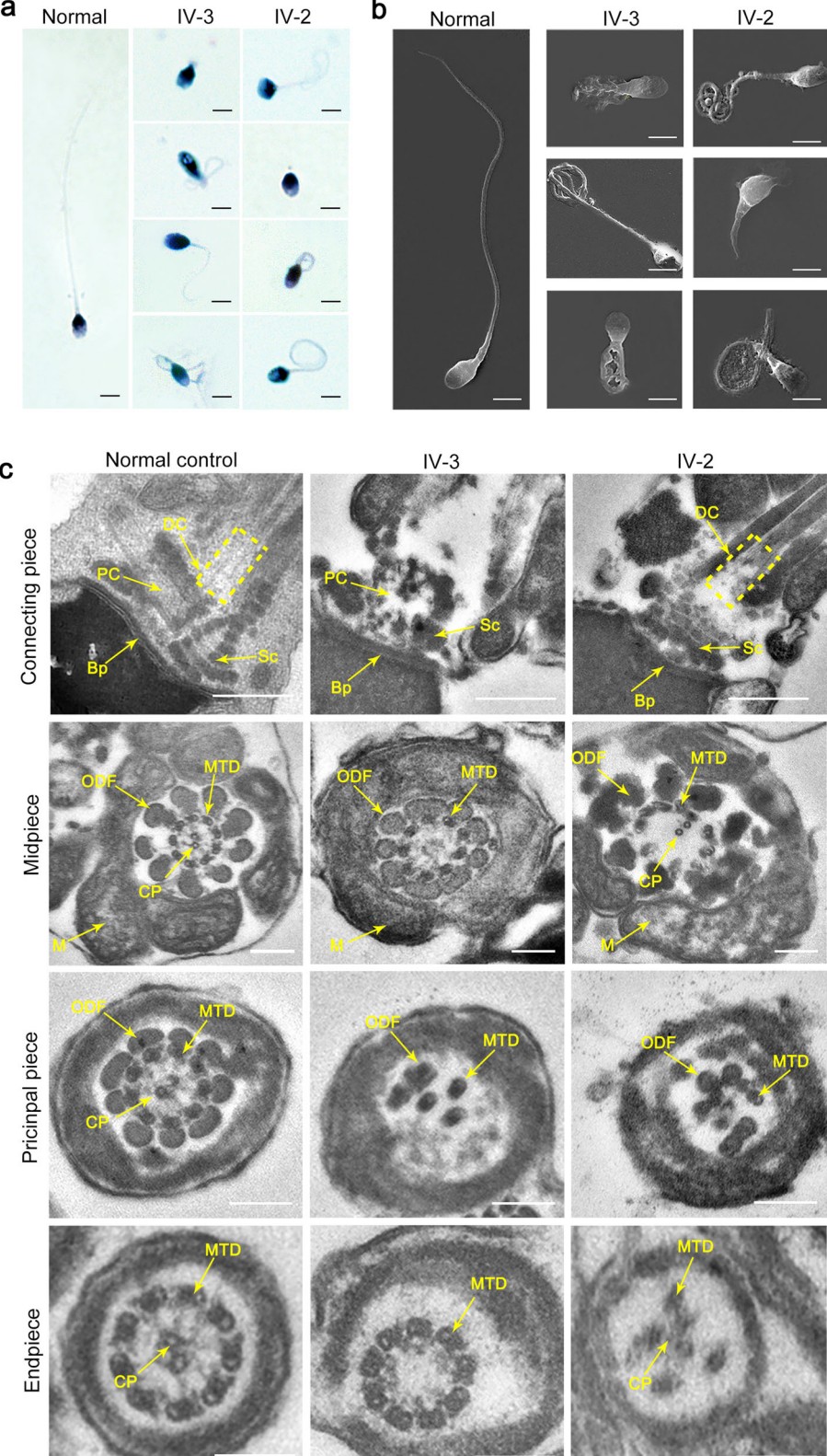

**Fig. 1 Morphological and ultrastructural defects in spermatozoa from two infertile siblings.** Abnormalities in sperm morphology observed in the two siblings by light microscopy (**a**) and scanning electron microscopy (SEM) (**b**) (scale bars, 5 μm). **c** The aberrations in sperm ultrastructure detected in the two affected individuals. The missing or incomplete PC, DC and Sc were found in the connecting piece, and the disorganized arrangements or absence of CPs, MTDs, and ODFs were observed in the patients' sperm flagella. Dotted box denotes DC. PC proximal centriole, DC distal centriole, Sc segmented column, Bp basal plate, CP central-pair microtubules, ODF outer dense fibers, MTD peripheral microtubule doublets, M mitochondria, IV-3 the proband, IV-2 the affected sibling (scale bars, 100 nm).

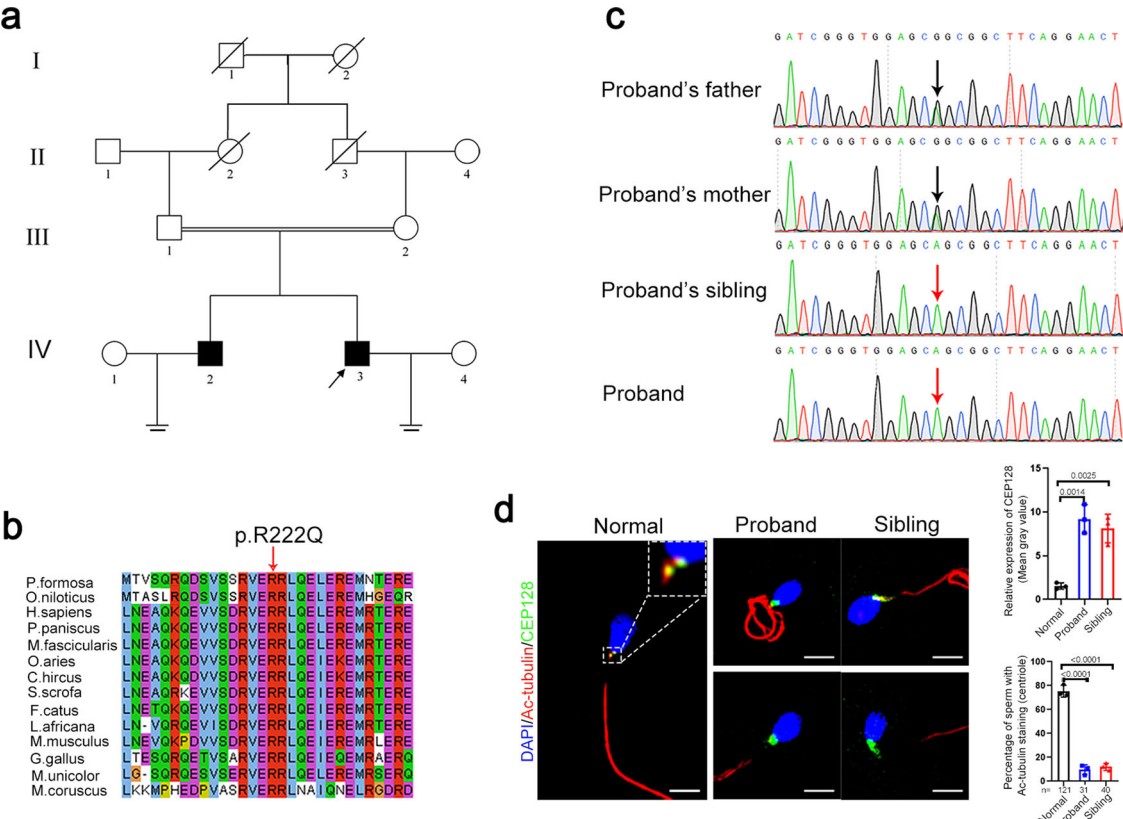

**Fig. 2 Identification of a homozygous variant in *CEP128* in two infertile siblings with cryptozoospermia from a consanguineous family. a** Pedigree of the consanguineous family with two infertile males (IV-2 and IV-3), with the arrow pointing to the proband. **b** Site of the *CEP128* missense variant is evolutionarily conserved among various species. **c** The affected males carry the homozygous missense variant in *CEP128*; the homozygotes are indicated by red arrows and the heterozygotes by black arrows. **d** CEP128 was localized to both PC and DC in the control spermatozoa. Dotted box denotes the colocalization of CEP128 with PC and DC. Increased abundances of CEP128 protein were observed in the sperm neck of the two siblings harboring a homozygous *CEP128* variant. Quantification of anti-CEP128 staining density in spermatozoa from control and patients. Quantification of the number of sperm with anti-Tubulin staining in the control and patients. Three independent experiments were performed. *n* the number of spermatozoa analyzed, PC proximal centriole, DC distal centriole. (Two-sided Student's *t* test; error bars, mean ± SEM; green, CEP128; red, Ac-Tubulin; blue, DAPI; scale bars, 5 μm). The *p* values are indicated in the graphs. Source data are provided as a Source Data file.

(WT)-*CEP128* and Flag-*CEP128*^R222Q plasmids and transfected them into 293T cells. The western blot results showed considerably increased levels of CEP128 protein in cells overexpressing the Flag-*CEP128*^R222Q plasmid compared to cells transfected with the Flag-WT-*CEP128* plasmid (Fig. 3a). This finding was consistent with the observation that CEP128 signals were dramatically increased in the sperm from the patients (Fig. 2d).

According to previous reports, CEP128 is involved in the cilia growth of hRPE-1 cells[15–17]. Therefore, we investigated the effect of the *CEP128*^R222Q variant on primary cilia development in hRPE-1 cells. EGFP-WT-*CEP128* and EGFP-*CEP128*^R222Q plasmids were generated and separately overexpressed in hRPE1 cells. The immunofluorescence staining results showed that short cilia were detected in cells transfected with EGFP-WT-*CEP128* plasmid and the nearly absent cilia was observed in cells with EGFP-*CEP128*^R222Q plasmid compared to the negative control (NC) cells in which endogenous CEP128 was located in the basal body of the cilia (Fig. 3b). Our findings were consistent with the previous suggestions that CEP128 overexpression could suppress ciliation in serum-starved hRPE-1 cells[15–17]. Additionally, compared to NC cells, most of the cells overexpressing the EGFP-*CEP128*^R222Q plasmid showed abnormal nuclear morphology, such as the smaller nucleus or nuclear fragmentation (Fig. 3b).

To understand how an elevation in CEP128 protein abundance gives rise to abnormalities in cilia development in hRPE-1 cells, we examined the expression of several well-established centrosomal proteins in cells transfected with the WT-*CEP128* and *CEP128*^R222Q plasmids as well as NC cells. As expected, the decreased protein levels of ninein (NIN), centriolin and CEP170 were detected in cells overexpressing WT-*CEP128* compared to the control, and the most obvious reduction was observed in the *CEP128*^R222Q group (Fig. 3c). Remarkably, we revealed that CEP128 transcriptionally downregulated the expression of NIN, centriolin and CEP170 (Fig. 3d). Next, GR-β, YY1 and CEBPβ were predicted to be the transcription factors regulating *NIN*, *centriolin* and *CEP170*, respectively, with the UCSC Genome Browser, JASPAR and ENCODE databases (Supplementary Table 2). Furthermore, the western blot results showed that the expression of GR-β, YY1 and CEBPβ was lower in cells transfected with the WT-*CEP128* and *CEP128*^R222Q plasmids than in NC cells (Fig. 3e). These data provide evidence to support the negative influence of *CEP128*^R222Q variant on primary cilia development in hRPE-1 cells.

**Exploration of the specific expression pattern of CEP128 in mouse testis.** The function of CEP128 in reproductive biology has remained entirely unknown since the protein was identified. In this study, we detected a functional homozygous missense

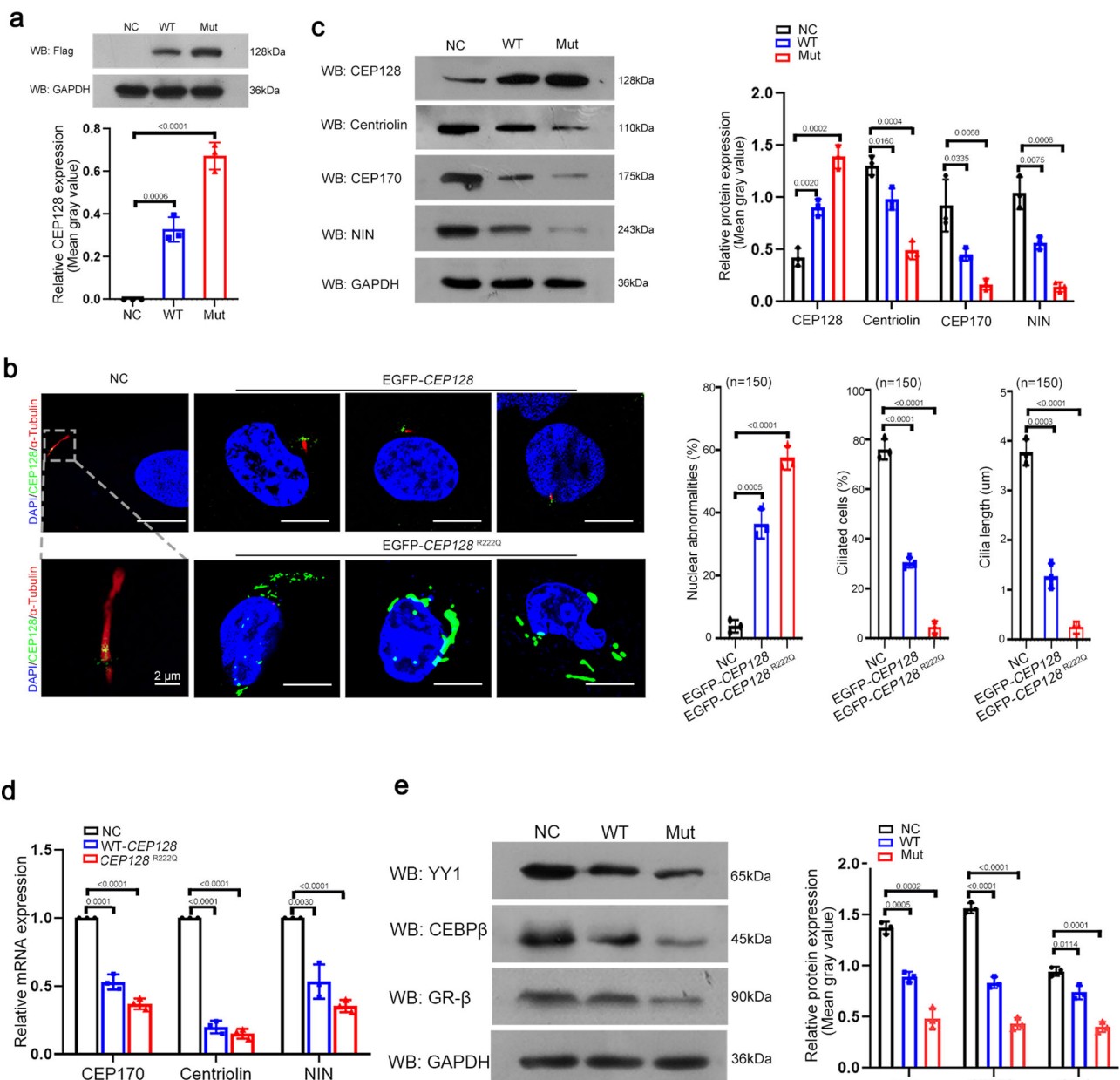

**Fig. 3 Functional analysis of the *CEP128*<sup>R222Q</sup> variant in hRPE1 cells. a** Western blot analysis showed an increased abundance of CEP128 protein with the *CEP128*<sup>R222Q</sup> variant. NC (negative control), cells transfected with negative control pENTER-Flag plasmid. Three independent experiments were performed. (Two-sided Student's *t* test; error bars, mean ± SEM). The *p* values are indicated in the graphs. **b** Overexpression of *CEP128*<sup>R222Q</sup> compromised ciliary growth and disrupted cellular nuclear morphology. Three independent experiments were performed. Dotted box denoted the basal body and cilia. *n* the number of cells analyzed. NC (negative control), cells transfected with negative control pENTER-Flag plasmid. (Two-sided Student's *t* test; error bars, mean ± SEM; green, CEP128; red, Ac-Tubulin; blue, DAPI; scale bars, 10 μm). The *p* values are indicated in the graphs. **c** Decreased levels of NIN, centriolin, and CEP170 were detected in the cells transfected with WT-*CEP128* or *CEP128*<sup>R222Q</sup> when compared to the NC by western blot analysis. NC (negative control), cells transfected with negative control pENTER-Flag plasmid. Three independent experiments were performed. (Two-sided Student's *t* test; error bars, mean ± SEM). The *p* values are indicated in the graphs. WT-*CEP128* and *CEP128*<sup>R222Q</sup> downregulated the mRNA levels for *NIN*, *centriolin*, and *CEP170* (**d**) by suppressing the expression of their transcription factors (**e**). NC (negative control), cells transfected with negative control pENTER-Flag plasmid. Three independent experiments were performed. (Two-sided Student's *t* test; error bars, mean ± SEM). The *p* values are indicated in the graphs. Source data are provided as a Source Data file.

variation in two infertile siblings; therefore, we hypothesized that CEP128 might play a pivotal role in spermatogenic process. Importantly, we found that CEP128 is predominantly expressed in the mouse testis (Supplementary Fig. 2a). Furthermore, we investigated the temporal expression of CEP128 in mouse testicular tissues obtained on different postnatal days and observed that CEP128 began to be obviously expressed on postnatal day 7,

reached peak expression on postnatal day 35, and then maintained a stable expression level thereafter (Supplementary Fig. 2b). Immunofluorescence staining revealed that CEP128 was detectable in the centrioles or their vicinity of various germ cells, except the steps 15-16 and the mature sperm (Supplementary Fig. 2c). Therefore, CEP128 might be a candidate molecule that contributes to sperm development involved in male fertility.

**Deficiency of *Cep128* results in infertility in male mice.** To determine the role of CEP128 during spermatogenesis, we generated *Cep128*-KO mice using CRISPR-Cas9 technology (Supplementary Fig. 3a). The successful establishment of the *Cep128*-KO mice was confirmed by Sanger sequencing, quantitative PCR (qPCR) and western blot (Supplementary Fig. 3b–d). The KO mice were viable and indistinguishable in appearance from the WT mice. Mating tests showed that the female KO mice were fertile (Supplementary Fig. 3e), and further haematoxylin-eosin (H&E) staining of ovarian tissue showed normal oocyte development (Supplementary Fig. 3f), indicating that *Cep128* does not play a role in female reproduction. As expected, the male homozygous KO mice were sterile after the mating period (Supplementary Fig. 3e). Compared to WT mice, the germ cells in KO mice were sparsely arranged during different spermatogenesis stages, as indicated by H&E staining (Fig. 4a). Using STA-PUT velocity sedimentation to separate spermatogenic cell types, we further defined the numbers of spermatocytes, round spermatids, and elongating/elongated spermatids were significantly reduced in testes of KO mice (Fig. 4b), which was in line with the enlarged image of testis cross-sections (Fig. 4c). Consequently, the sperm cells dramatically declined in the different segments of the KO mouse epididymis compared to those of WT mice (Fig. 4d). The computer-assisted sperm analysis (CASA) was carried out to exhaustively examine the sperm quality of the KO mice. Of note, reduced sperm motility, abnormal sperm morphology and decreased sperm count were observed in the KO mice compared to WT mice (Supplementary Movies 1 and 2). The detailed differences were presented in Table 1. Notably, the sperm morphological deformities in KO males were further demonstrated by light microscopy and were revealed to consist mainly of decapitated and decaudated defects; however, other morphological abnormalities could also be found (Fig. 5a and Supplementary Fig. 4a). To delineate the anomalies of the sperm-connecting piece in KO mice, we performed TEM on testicular cross-sections as well as epididymal spermatozoa from KO and WT mice. Only partially formed or severely disorganized PC was observed in step 9–14 spermatids from KO mice, while the formation of Sc and Bp was normal (Fig. 6c and Supplementary Fig. 4b). To determine whether the reduced motility and abnormal tails of spermatozoa resulted from defective flagellar ultrastructure in KO mice, we further applied TEM to the sperm flagella. As expected, flagellar defects were markedly increased in KO mice: an atypical "9 + 0" or "9 + 1" arrangement of axonemal microtubules was found in most of the midpiece; absent or disorganized CPs were detected in the principal piece; and in the end piece, the disappeared or disordered MTDs were apparent (Fig. 5b and Supplementary Fig. 4c). However, no obvious defects in the length and ultrastructure of cilia from the lungs, tracheas, eyes, and brains were detected in the KO mice by morphological analysis (Supplementary Fig. 5a–d). These results suggest that *Cep128* plays a crucial role in spermatogenesis, and loss of CEP128 expression can cause male infertility in mice.

**Abnormal spermatid development in homozygous *CEP128*^R222Q KI male mice.** Since the pivotal role of *Cep128* in spermatogenesis was unveiled in KO mice, we further used CRISPR-Cas9 to generate the mice encoding orthologous variant of *CEP128*^R222Q detected in humans to confirm the negative effect of this variant in mice (Supplementary Fig. 3a). PCR sequencing was used to verify the success of *Cep128*-KI mice (Supplementary Fig. 3b). Western blot validated the increased expression of CEP128 in testes of homozygous KI mice (Supplementary Fig. 3d). The homozygous KI mice displayed normal development, and natural oogenesis was found in homozygous KI females (Supplementary Fig. 3f).

However, WT females produced no pups after breeding with homozygous KI males for several months (Supplementary Fig. 3e). The impaired spermatogenesis observed in homozygous KO mice was also discernible in the homozygous KI mice (Fig. 4a–d). CASA was then used to analyze the epididymal sperm parameters of homozygous KI mice. Greatly reduced sperm count and motility were found in homozygous KI mice compared to those in WT mice (Supplementary Movie 3 and Table 1). Most of the immotile sperm cells showed hairpin bending between the midpiece and the principal piece (Fig. 6a and Supplementary Fig. 4d). Considering that the annulus, a Septin cytoskeletal structure located between the midpiece and principal piece regions of the sperm tail, is involved in bending defects[18,19], we labeled the annular protein Septin 4 in sperm of homozygous KI mice by immunofluorescence staining and found that Septin 4 expression was similar in homozygous KI mice and WT mice (Supplementary Fig. 6a). As shown in the enlarged images, the intact annulus was observed in homozygous KI males by SEM (Supplementary Fig. 6b). Thus, we supposed this kind of devoid morphotype might result from the abnormal arrangements of sperm axoneme. To address this supposition, TEM was applied to investigate the sperm ultrastructure of homozygous KI mice. Intriguingly, the principal piece and the end piece exhibited a classic "9 + 2" axonemal configuration. However, in most of the midpiece, the number of axonemes was normal, while MTDs 1–3 and ODFs 1–3 showed an irregular arrangement (Fig. 6b and Supplementary Fig. 4e). Notably, the abnormalities of the spermatozoa head-neck ultrastructure were also observed in homozygous KI mice in different stages of spermiogenesis (Fig. 6c and Supplementary Fig. 4b). The Sc either did not develop or was partially formed, and the nine centriole triplets were irregularly arranged or incomplete. Similarly, natural cilium formation was exhibited in the lungs, tracheas, eyes, and brains of homozygous KI mice (Supplementary Fig. 5a–d). In addition, we checked the normal spermatogenesis in heterozygous KI males (Supplementary Fig. 7, Supplementary Table 3, and Supplementary Movie 4). These data thus indicate the pathogenicity of the homozygous *CEP128*^R222Q variant in spermatogenesis in mice.

**Altered expression of reproduction-related genes in homozygous *Cep128*-mutated mouse models.** To determine the molecular mechanism of the involvement of CEP128 in male fertility, we employed the proteomics approach to examine the testes of KO, KI and WT adult mice. Among the 399 differentially expressed proteins identified by tandem mass tag (TMT) quantification (Fig. 7a and Supplementary Dataset 2), proteins required for energy metabolism and reproduction were downregulated in homozygous KO and KI mice compared to WT mice, while proapoptotic proteins were found at higher levels in KO (Fig. 7b) and KI mice (Fig. 7c). We focused on five key proteins, including Wbp2nl, Rcbtb2, Prss55, Crisp1 and Defb22, for further study according to previous suggestions that ablation of these proteins in mice could impair male fertility[20–27]. Substantially reduced levels of Prss55 and Crisp1 in KO mouse testes and significantly diminished levels of Wbp2nl, Rcbtb2, Prss55 and Crisp1 in homozygous KI mouse testes were further confirmed by immunofluorescence staining (Supplementary Fig. 8a, b). Decreased *Defb22* expression was detected in both homozygous KO and KI mice by qPCR (which was performed because of a lack of commercial antibody) (Supplementary Fig. 9a). Via co-immunoprecipitation assay, we further found that CEP128 could bind to Wbp2nl, Rcbtb2, Prss55 and Crisp1, and the increased ubiquitination-mediated degradation of Wbp2nl, Rcbtb2, Prss55 and Crisp1 was detected in homozygous KO and KI mice compared to WT mice (Supplementary Fig. 8c–h). Moreover, RNA-sequencing of testes separately revealed 1614 and 985 genes that were differentially expressed in homozygous KO and

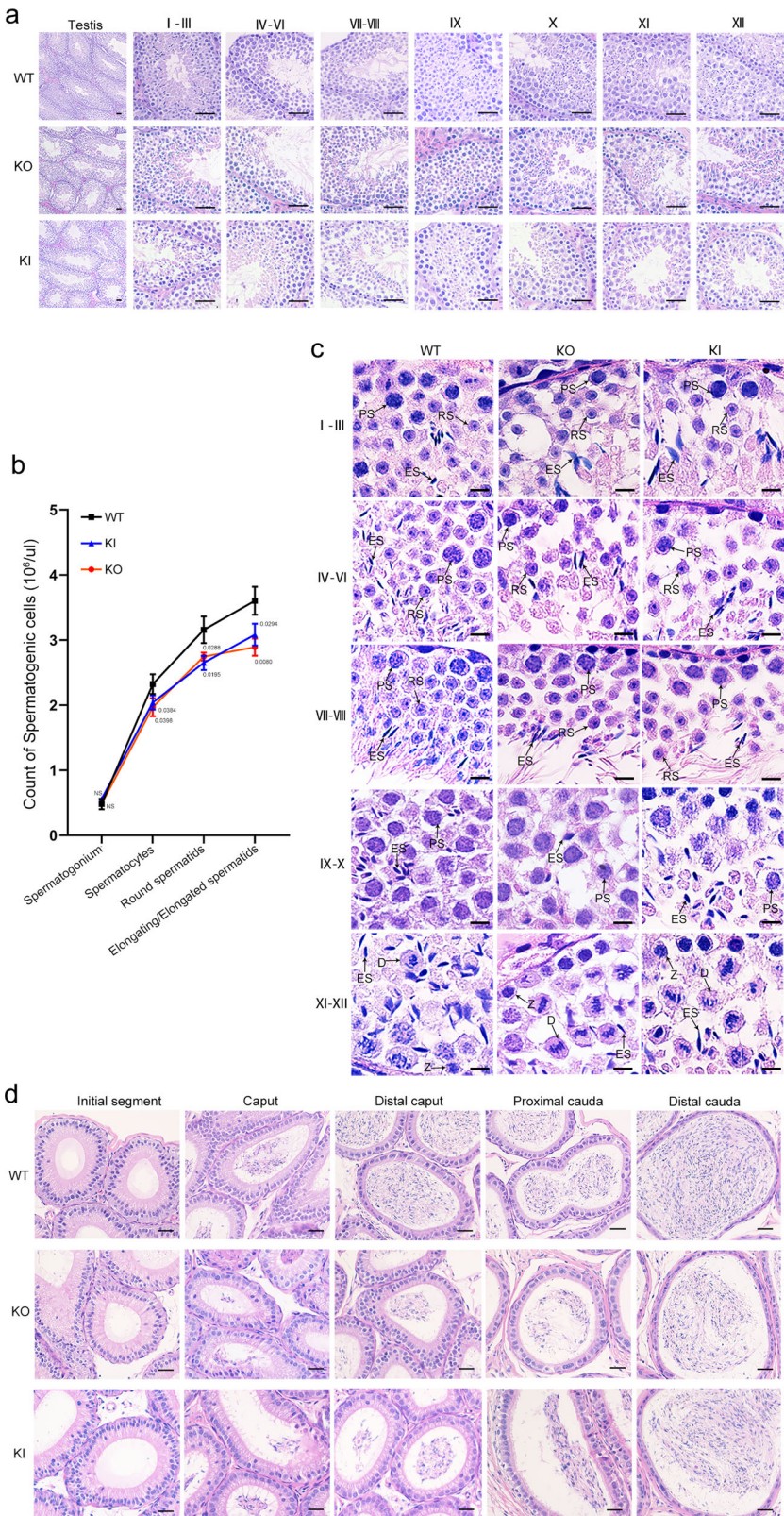

KI mice compared to WT mice (Fig. 7d, e and Supplementary Dataset 3). Decreased expression of genes involved in spermatogenesis, including sperm flagellar development and sperm production as well as in fertilization, was observed in KO mice (Fig. 7d)[28–44] and KI mice (Fig. 7e)[31,39–53] compared to WT mice. The expression of genes exclusively or predominantly expressed in testes was substantially diminished in homozygous KO and KI mice

(Fig. 7d, e). The qPCR assay confirmed the decreased expression of several key genes in KO and KI mice (Supplementary Fig. 9a–c).

We next employed a quantitative phosphoproteomics analysis on WT/KO/KI mouse testes to further decipher the potential molecular mechanisms governing CEP128 function in spermatogenesis, as CEP128 loss appeared to attenuate TGF-β/BMP-induced phosphorylation of multiple proteins that regulate cilium-associated

**Fig. 4 Impaired spermiogenesis in male homozygous *Cep128* KO and KI mice. a** H&E staining of the testicular sections from adult male homozygous *Cep128* KO and KI mice showed a decrease in germ cells at different stages of spermatogenesis ($n = 3$ biologically independent WT mice, KI mice or KO mice; scale bars, 125 μm). **b** Spermatocytes, round spermatids, and elongating/elongated spermatids were decreased in homozygous KO and KI mice as assessed by STA-PUT velocity sedimentation ($n = 3$ biologically independent WT mice, KI mice or KO mice; two-sided Student's *t* test; error bars, mean ± SEM). The *p* values are indicated in the graphs. **c** H&E staining confirmed the reduced spermatocytes, round spermatids, and elongating/elongated spermatids in KI and KO mice. PS primary spermatocyte, RS round spermatid, ES elongating/elongated spermatid ($n = 3$ biologically independent WT mice, KI mice or KO mice; scale bars, 20 μm). **d** Fewer spermatozoa were observed in individual fields of the epididymis in homozygous *Cep128* KO and KI mice relative the epididymides of WT mice ($n = 3$ biologically independent WT mice, KI mice or KO mice; scale bars, 125 μm). Source data are provided as a Source Data file.

**Table 1 Semen analysis using CASA in the mouse model of *Cep128*.**

|  | Adult male mice | | |
|---|---|---|---|
|  | **WT** | **KO** | **KI** |
| Semen parameters |  |  |  |
| Sperm concentration ($10^6$/ml)[a] | 27.80 ± 5.54 | 15.32 ± 1.43 | 11.28 ± 1.87 |
| Motility (%) | 56.60 ± 8.82 | 3.09 ± 0.90 | 2.87 ± 1.67 |
| Progressive motility (%) | 54.66 ± 11.30 | 1.82 ± 0.84 | 1.75 ± 1.11 |
| Sperm locomotion parameters |  |  |  |
| Curvilinear velocity (VCL) (μm/s) | 57.05 ± 7.20 | 10.16 ± 1.39 | 8.11 ± 2.28 |
| Straight-line velocity (VSL) (μm/s) | 17.37 ± 3.79 | 0.92 ± 0.30 | 0.34 ± 0.21 |
| Average path velocity (VAP) (μm/s) | 26.87 ± 2.56 | 3.00 ± 1.50 | 1.81 ± 0.82 |
| Amplitude of lateral head displacement (ALH) (μm) | 0.98 ± 0.16 | 0.21 ± 0.02 | 0.17 ± 0.04 |
| Linearity (LIN) | 0.37 ± 0.05 | 0.05 ± 0.02 | 0.04 ± 0.04 |
| Wobble (WOB = VAP/VCL) | 0.48 ± 0.10 | 0.29 ± 0.09 | 0.22 ± 0.05 |
| Straightness (STR = VSL/VAP) | 0.64 ± 0.09 | 0.33 ± 0.09 | 0.18 ± 0.06 |
| Beat-cross frequency (BCF) (Hz) | 6.29 ± 0.67 | 0.81 ± 0.33 | 0.35 ± 0.20 |

[a]Epididymides and vas deferens.

vesicle trafficking[54]. Among the differentially phosphorylated sites identified in our study (Supplementary Dataset 4), we found that TGF-β/BMP-signaling members (including Lefty2, Neo1, Fbn1, Rbl1, Gata4, and Trim33) exhibited greatly reduced phosphorylation in homozygous KI mice compared to WT mice (Supplementary Fig. 10). Notably, Rbl1, Gata4, and Trim33 were shown to be involved in the regulation of spermatogenesis (Fig. 7f); Rbl1 participates in the control of germ cell proliferation, differentiation and apoptosis[55,56]; Gata4 cKO mice showed decreases in the quantity and motility of sperm[57]; and the testicular expression pattern of Trim33 indicated a possible involvement of Trim33 in spermatogenesis[58]. In addition, the phosphorylations of several other non-TGF-β/BMP-signaling proteins that are associated with the spermatogenic process were also significantly diminished in homozygous KI mice—such as Fsip2, Cfap251, Akap3, Akap4, Cep131, and Odf2 (Supplementary Fig. 10)[12,59–63]. Investigators have not uncovered significant differences in the phosphorylation of TGF-β/BMP-signaling components in KO mice compared to WT mice, while some proteins involved in spermatogenesis exhibited obviously reduced phosphorylation in KOs vs. WT mice—including Fsip2, Cfap251, Prm1, Hfm1, Fam170b, Wipf3, and Top2a (Supplementary Fig. 11)[59,60,64–68]. To summarize, CEP128 is a regulator of normal reproductive functioning via its modulation of essential molecules involved in spermatogenesis and fertilization.

**Poor outcomes of intracytoplasmic sperm injection (ICSI) using the sperm from homozygous *Cep128*-mutated male mice and *CEP128*-associated patients.** ICSI cycles were attempted for the patients, and the written informed consents were obtained for the ICSI procedure. The proband' wife, who had normal basal hormone levels, underwent a long gonadotrophin-releasing hormone agonist protocol (Supplementary Table 4). Eight oocytes were retrieved, and four mature oocytes were microinjected; two were ultimately fertilized. Regrettably, only one embryo reached

the cleavage stage, and that embryo stopped developing. The basal hormone levels of the sibling's wife were also normal. She was subjected to a long protocol, and nine oocytes were retrieved (Supplementary Table 4). Finally, one of seven mature oocytes was fertilized by ICSI, but the oocyte failed to develop after reaching the cleavage stage. These results indicated that *CEP128*[R222Q] variant might be linked to a poor prognosis in regard to ICSI treatment. While the additional female risk factors of infertility should not be excluded, and more cases needs to be investigated to make clear the role of this mutation on ICSI outcomes.

Noticeably, the disappointing outcomes of ICSI were exhibited in the homozygous KI and KO male mice. After injection, pronuclei were observed in most of the embryos in the WT group, whereas the fertilization rates of the homozygous KI and KO groups were only 50% and 6%, respectively (Fig. 8). The percentage of 2-cell embryos was significantly lower in homozygous KI mice than in WT mice, and few 2-cell embryos were obtained in the KO mice (Fig. 8). Only 18% of the embryos from homozygous KI mice progressed from the 2-cell stage to the blastocyst stage, and an even smaller percentage of zygotes from KO mice reached the blastocyst stage (Fig. 8). The impaired fertilization observed in homozygous KO and KI mice might be associated with the reduced expression of Crisp1, Wbp2nl, Tmem95, Eqtn and Calr3, which are involved in gamete fusion and egg activation mediated by meiotic resumption and pronuclear development[22–25,41,47,48]. Taken together, these data demonstrate that *Cep128* plays specific biological roles in fertilization and early embryonic development in mice.

## Discussion
Male fertility entails the capacity of the seminiferous tubules to continuously produce normal, mature spermatozoa from differentiating spermatogonia. This exquisitely elaborate spermatogenic process is, however, easily disturbed by environmental

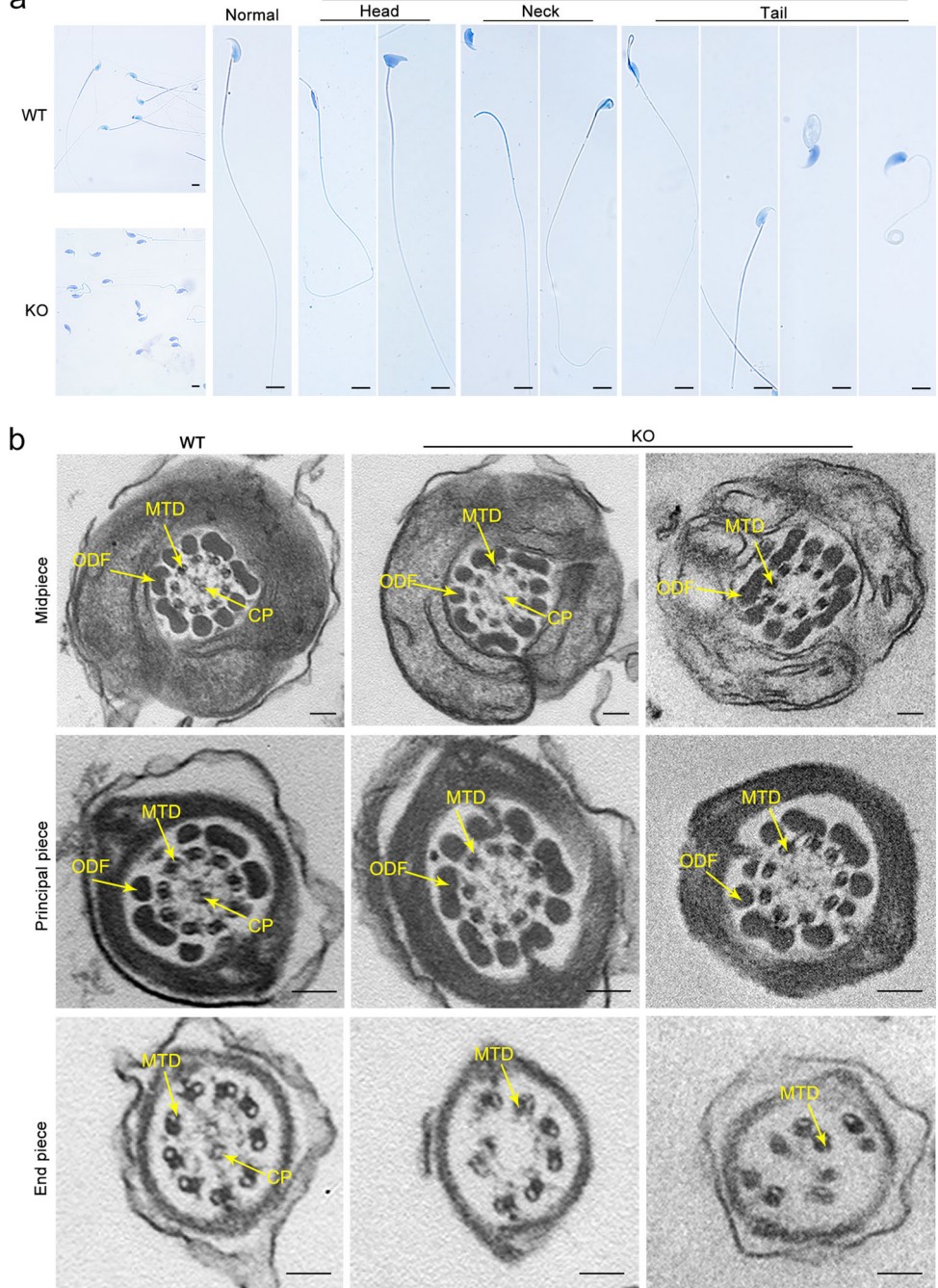

**Fig. 5 Anomalies in the morphology and ultrastructure of spermatozoa from male *Cep128* KO mice. a** Spermatozoa from male KO mice showed prominent deformities in the head, neck, and tail regions under light microscopy ($n = 3$ biologically independent WT mice or KO mice; scale bars, 20 μm). **b** The primary defect in the midpiece and principal pieces was the absence of the CP, while missing and misplaced MTDs were observed in the end piece. CP central-pair microtubules, MTD peripheral microtubule doublets. ($n = 3$ biologically independent WT mice or KO mice; scale bars, 150 nm).

factors as well as for intrinsic pathogenic reasons. Similar to what has been found for other disorders, genetics appears to play a predominant role in spermatogenic defects. Using a variety of genomic techniques, the genes involved in male infertility have now been largely identified, while the genetic causes of numerous spermatogenic impairments still remain decidedly unclear.

In this study, we identified a homozygous missense variant of *CEP128* in two siblings with infertility characterized by an extremely low sperm count and complete sperm morphological anomalies. Intriguingly, mice harboring the orthologous variant of *CEP128*[R222Q] represented the diminish in sperm count and motility,

and *Cep128*-deficient male mice showed more severe abnormalities in sperm parameters, indicating the pivotal role of CEP128 in male reproduction. We eventually uncovered the mechanism in which CEP128 regulates the expression of key molecules including Wbp2nl, Rcbtb2, Prss55, Crisp1, Defb22, Sun5, Tssk4, p-Rbl1, p-Gata4 and p-Trim33, which are involved in the reproductive process; and variants that cause anomalous expression of the CEP128 protein inhibit the functions of these crucial molecules, further leading to aberrant sperm development and fertilization (Fig. 9). Collectively, these findings comprehensively elucidate the essential function of CEP128 in the reproductive process.

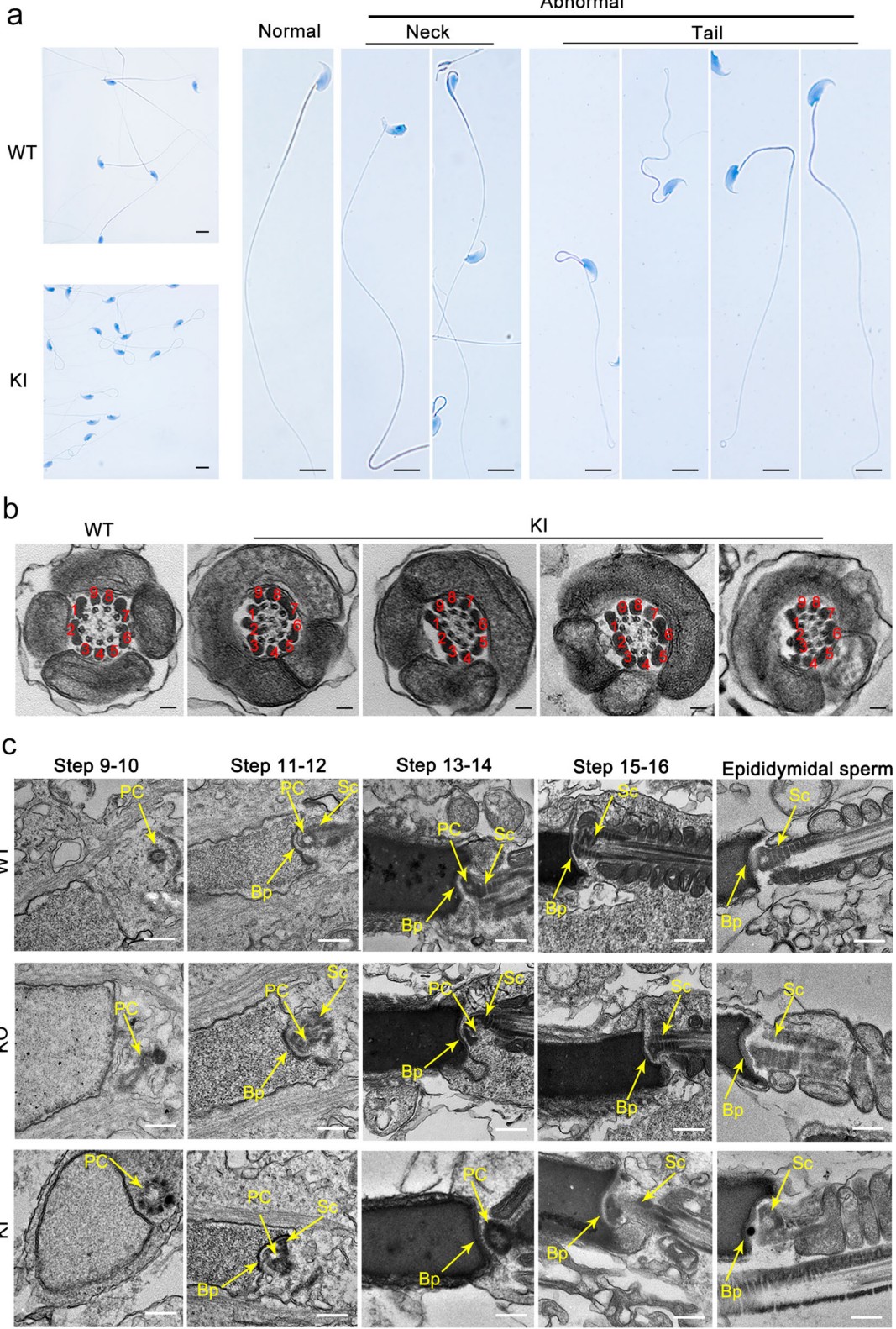

**Fig. 6 Multiple defects are observed in spermatozoa from male homozygous *Cep128* KI mice. a** Male homozygous KI mice showed hyperplasia of sperm flagella (*n* = 3 biologically independent WT mice or KI mice; scale bars, 20 μm). **b** Dislocated arrangements of MTDs 1–3 and ODFs 1–3 were exhibited in the majority of the midpieces. ODF outer dense fibers, MTD peripheral microtubule doublets. (*n* = 3 biologically independent WT mice or KI mice; scale bars, 150 nm). **c** Impaired development of the connecting piece during spermiogenesis in male homozygous *Cep128* KO and KI mice. In spermatid steps 9–14, PCs with incomplete structure were present in both male homozygous KO and KI mice. Signs of defective Sc formation were occasionally visible in spermatids from male homozygous KI mice, but not in KO mice. In contrast, the Bp was almost normal in both male homozygous KO and KI mice. PC proximal centriole, Sc segmented column, Bp basal plate. (*n* = 3 biologically independent WT mice, KO mice or KI mice; scale bars, 500 nm).

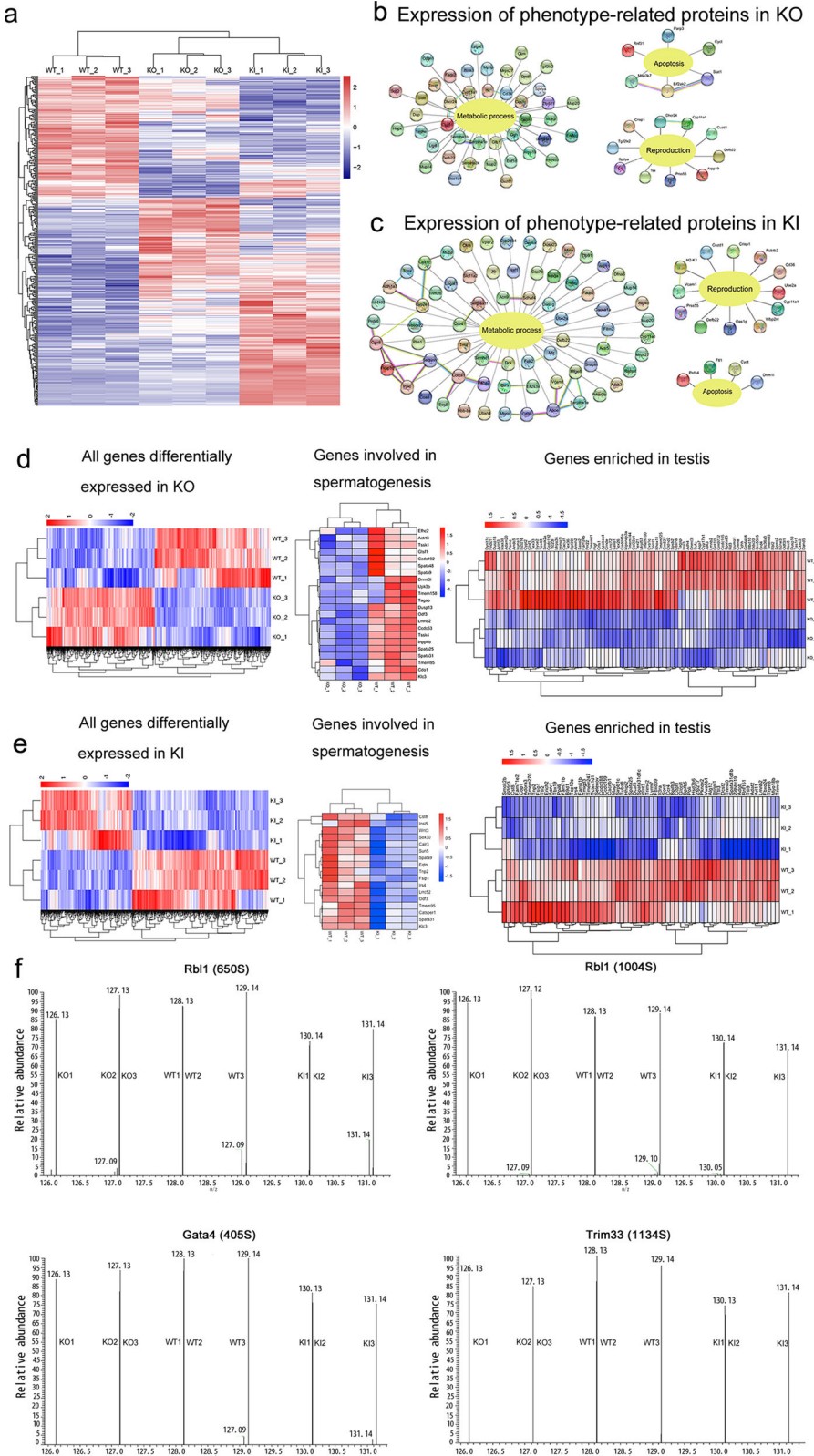

Although the highest allele frequencies of the *CEP128* p.R222Q variant was reported in the Japanese (0.5–2.34% according to the 1000 Genomes Project and 8.3KJPN project)[69] and Korean populations (1.13% according to the KRGDB project) [https://www.ncbi.nlm.nih.gov/bioproject/PRJNA589833], this variant is rare or absent in the remaining populations of the world. For example, *CEP128* p.R222Q was not detected in our 1000 normal Chinese control males from Sichuan, China. Moreover, the general population genome database (1000 Genomes Project) showed the frequency of this variant in Southern Han Chinese (CHS) is 0.0048, and it is absent in Han Chinese in Beijing (CHB) and African, European, and American populations. We speculated that the specific distributions of the *CEP128* p.R222Q variant in the Japanese and Korean populations might be exceptional accordingly.

**Fig. 7 The functional mechanisms underlying CEP128 regulation of male reproduction. a** Heatmap of the results of proteomic analyses of testes from adult male WT, homozygous *Cep128* KI, and KO mice. Levels of proteins involved in energy metabolism and reproduction were significantly reduced, and proapoptotic proteins were increased in male homozygous KO (**b**) and homozygous KI mice (**c**) when compared to male WT mice. **d** Heatmap of genes differentially expressed between male WT and KO mice as determined by RNA-sequencing. A diminution in the expression of genes associated with spermatogenesis and those that were enriched in testes were decreased in male KOs compared to WT males. **e** Heatmap of genes differentially expressed between male WT and homozygous KI mice by RNA-sequencing. The genes associated with spermatogenesis and enriched in the testes were decreased in male homozygous KI mice compared to WT males. **f** The differential MS/MS spectra of p-Rbl1, p-Gata4, and p-Trim33 as detected in the testes of homozygous KI mice. ($n = 3$ biologically independent WT mice, KO mice or KI mice).

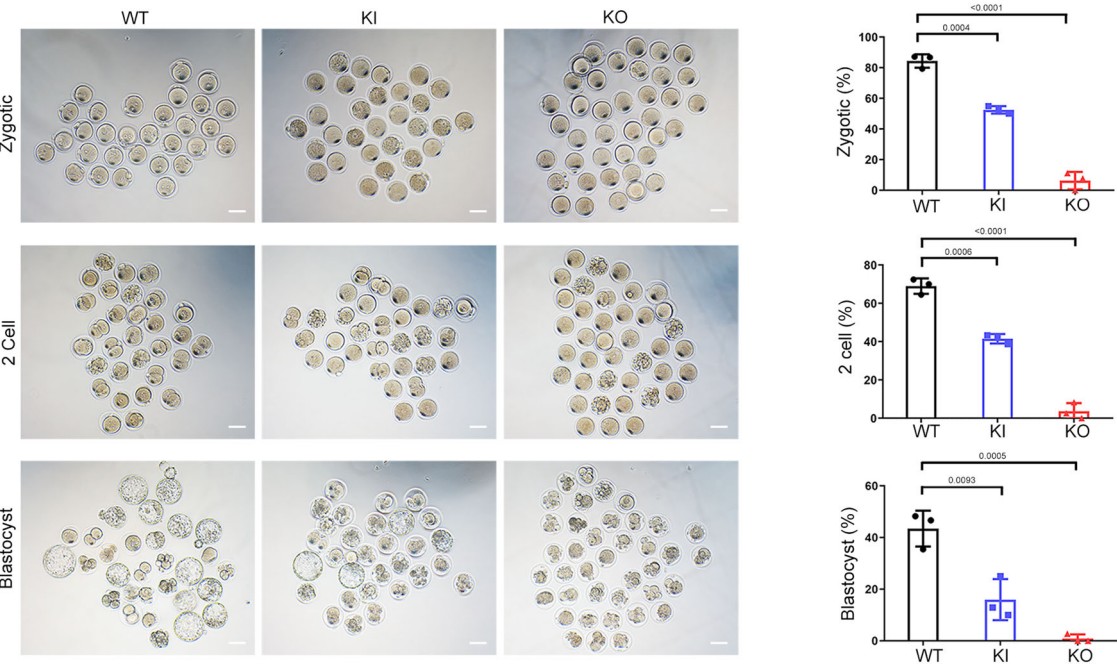

**Fig. 8 Poor prognosis after ICSI treatment using the sperm from homozygous *Cep128* KO and KI mice.** We observed significantly decreased percentages of one-cell zygotes, two-cell embryos, and blastocysts in the homozygous KI group, and the KO group showed almost total fertilization failure ($n = 3$ biologically independent WT mice, KI mice or KO mice; scale bars, 100 μm; two-sided Student's *t* test; error bars, mean ± SEM). The *p* values are indicated in the graphs. Source data are provided as a Source Data file.

As we known, male infertility is a common disease with a high incidence of ~7% in human populations[2]. Therefore, it may be reasonable to observe the allele frequency of ~1% for a recessive pathogenic allele of male infertility (the frequency of the homozygote is ~0.01% in populations). In addition, a confirmed pathogenic variant can lead to the phenotypic variance of between populations. For instance, the gr/gr deletion on the human Y chromosome is a significant risk factor for spermatogenic failure and this deletion is almost absent in the Dutch and some other Caucasian populations[70]. However, the prevalence of the gr/gr deletion varies dramatically across different populations; and the highest frequency of the gr/gr deletion is in Japan (33.7%)[71,72]. These higher allele frequencies of disease-associated variants than expected in some specific populations may reflect the complex genetic mechanisms underlying the genotype-phenotype correlation, such as the impacts of genetic background and phenotype-modifier genes. Reasonably, it should be an exception that the higher distributions of the *CEP128* p.R222Q variant in the Japanese and Korean population, and that might be due to their specific genetic background and/or the impacts of phenotype-modifier genes. Certainly, we identified this variant only in two siblings, so more cases with male infertility need to be investigated in the future. Detecting pathogenic mutations of *CEP128* in unrelated infertile patients would provide strong evidence to confirm the causative role of *CEP128* deficiency in human spermatogenesis.

CEP128, an important subdistal appendage protein, has been suggested to mediate primary ciliogenesis via interaction with ODF2 to promote centriolin, NIN, and the NIN group members localizing to the centriole subdistal appendages in hRPE-1 cells[13,14]. Previous data pointed out that CEP128 overexpression can suppress ciliation, and a lack of CEP128 in serum-starved hRPE-1 cells resulted in elevated levels of ciliation[15–17]. Consistent with these findings, in our study, the *CEP128*[R222Q] variation led to the abnormally high levels of CEP128 further hampered cilia growth in cultured cells. However, in our study we observed no significant differences in the distribution or length of cilia in the lungs, tracheas, eyes, or brains between homozygous *Cep128* KO/KI mice and WT mice. In addition, both homozygous *Cep128* KO and KI mice showed a typical ciliary ultrastructure in these tissues. Our patients did not present any chronic disease-associated defects in cilia in other organs except for testis. Similarly, loss of CEP131—a centrosomal protein localized to centriolar satellites—leads to a robust reduction in ciliogenesis in mouse fibroblasts; in contrast, *Cep131*-null mice demonstrated no discernible ciliary phenotype while showing defects in both the manchette and the flagella of sperm, resulting in male infertility[12]. Thus, we hypothesized that ciliary development in other organs was regulated or compensated by other centrosomal proteins so as to allow ciliogenesis to proceed despite the lack of CEP128. Our findings unveil that CEP128 is an essential modulator of ciliogenesis in testes instead of other tissues.

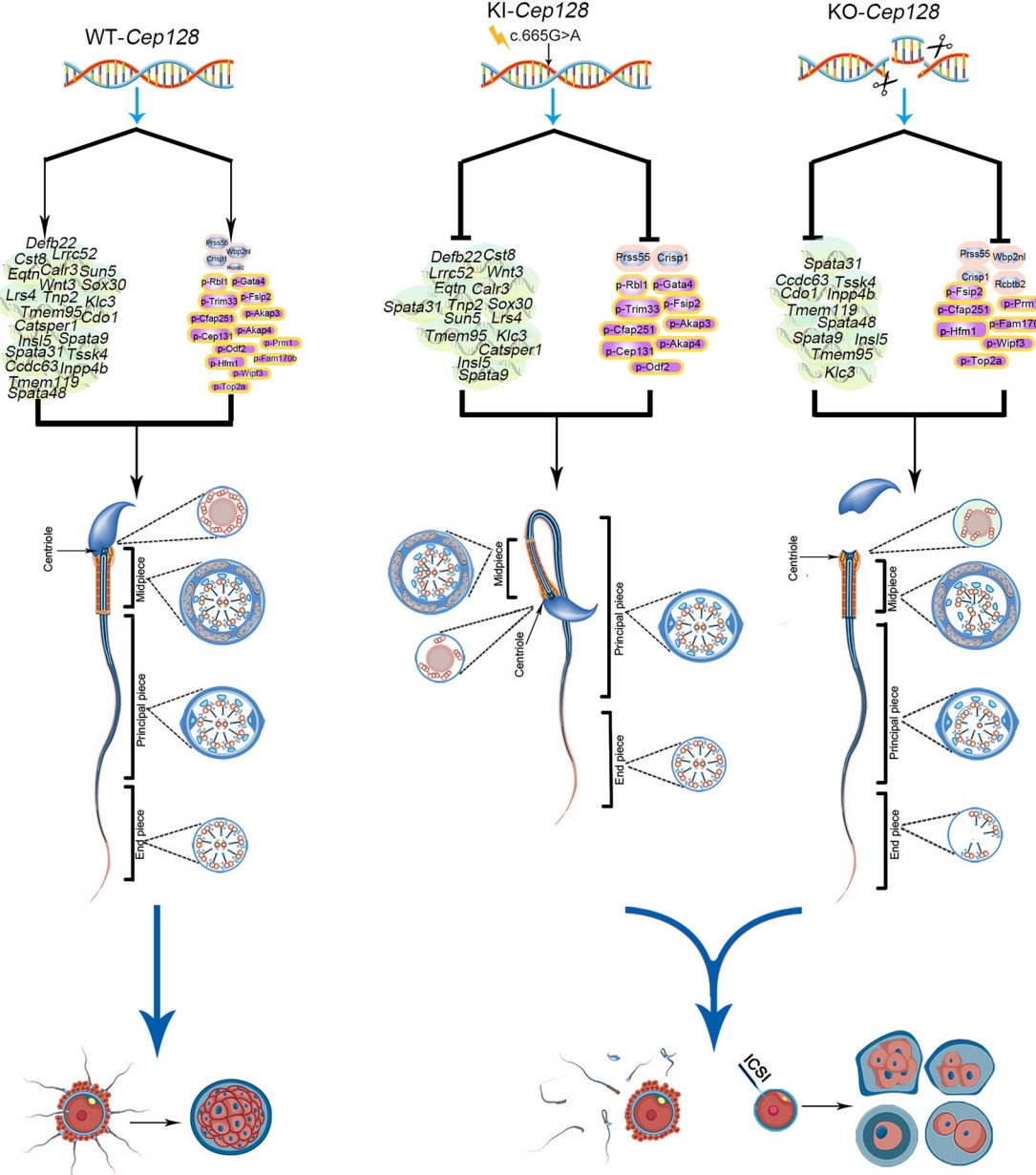

**Fig. 9 Proposed model of CEP128 modulation of male reproduction.** The imbalance observed in CEP128 levels included increased CEP128 abundance or loss of CEP128 protein, which suppressed the normal functioning of genes/proteins essential for sperm production and morphology as well as fertilizing capability. Normal sperm eventually failed to form, impairing typical reproductive function.

The centrosome is an evolutionarily conserved cellular component essential for the reproductive process. During sperm development, the centrosome plays a particular role in maintaining the head-tail connection and extends to form the axonemal microtubules of the flagella[73]. During fertilization in humans and most non-murine mammals, centrosomes are required for zygote division through the spindle poles and further mediate chromatid separation between daughter cells[74]. Therefore, the pathogenic variations in the genes encoding centrosome-associated proteins are naturally responsible for centrosome dysfunction and ultimately for infertility. Regretfully, exploration of the involvement of centrosomal proteins in reproduction has been limited. A previous report described an infertile patient carrying a homozygous variant (p.D455V) in *CEP135* who showed the MMAF phenotype and for whom ICSI treatment was unsuccessful because of a lack of CEP135 in the PCs[9]. Loss-of-

function variants in *CEP112* have been reported to severely disrupt head-tail attachment and lead to acephalic spermatozoa[10]. Intriguingly, in our study, two infertile siblings harboring the *CEP128* variant presented severe defects in sperm counts, morphology and motility, and ICSI failed in these patients might result from the aberrant centrioles exhibited in almost all the spermatozoa. Most importantly, homozygous *Cep128* KO and KI mice showed the impaired spermatogenesis. The relationships between CEP128 and diseases have not been characterized since it was identified, except in one study that suggested that CEP128 plays a crucial role in the pathogenesis of Graves' disease[75]. Therefore, the proteomics approach, RNA-sequencing and phosphoproteomics analysis were employed on mice testes to elucidate the potential mechanism of CEP128 involvement in male infertility. According to bioinformatics analysis and the literature, genes/proteins that play essential roles in sperm

production and fertilization were substantially downregulated in homozygous KO and KI mice; the diminished phosphorylation of the TGF-β/BMP-signaling members Rbl1, Gata4, and Trim33 (proteins that are correlated with reproductive processes) in homozygous KI mice suggesting that CEP128 participates in spermatogenesis via regulating the phosphorylation of downstream components of the TGF-β/BMP-signaling pathway. In zebrafish, the loss of CEP128 impaired organ development by reducing phosphorylation of TGF-β/BMP signaling[54]. Collectively, our data suggest that phosphorylation of TGF-β/BMP-signaling members constitutes a pivotal target for CEP128 functioning in various biological processes. Our findings reveal the functional mechanism by which CEP128 regulates male reproduction and provide more guidance for future research.

In summary, the current findings uncovered CEP128 plays a critical role in male reproduction. Our study substantiates that CEP128 as the important centrosomal protein identified to be associated with sperm formation and zygote development in mice. The elucidated mechanism underlying the role of CEP128 in the male reproductive process provides insights into the function of centrosomal proteins in male fertility. Further genetic screening of *CEP128* in additional infertile men may provide more information to explore the complex causes of male sterility. Overall, this study expands our knowledge of the relationship between centrosomal proteins and disease, opening avenues for disease treatment strategies.

## Methods

**Study participants**. All the researches on human subjects obtained ethical approval given by the Ethical Review Board of West China Second University Hospital, and each subject signed informed consent. The brothers with primary infertility and their parents were recruited from the West China Second University Hospital, Sichuan University. Furthermore, 1000 healthy men (25–35 years old) who had undergone medical check-ups without evidence of any infertility and had fathered at least one child were enrolled as normal controls.

**Whole-exome sequencing and Sanger sequencing**. Genomic DNA was isolated from peripheral blood samples of the patients using the QIAamp DNA Blood Mini Kit (51104, QIAGEN). Exome captures were performed with the Agilent SureSelect Human All Exon V6 Kit (G9706K, Agilent Technologies). Sequencing was then carried out on the Illumina HiSeq X system according to the manufacturer's instructions. The average sequencing depth on targets was ~107×, and the ratio of the target fraction covered at minimum 10× was 99.5%. Sequence reads were aligned to the human genome reference assembly (GRCh37/hg19) by the Burrows Wheeler Aligner software. ANNOVAR software was utilized for functional annotation with information from a variety of databases including dbSNP, 1000 Genomes Project, ExAC, Gene Ontology (GO), KEGG and HGMD. The Genome Analysis Toolkit (GATK 3.7) was employed to identify and quality-filter the variants.

The *CEP128* variant of c.665G > A (p.R222Q) identified by whole-exome sequencing was validated by Sanger sequencing in the patients and their unaffected parents. The primers of PCR: Forward, 5′-GGATCCAGGTGTCGATTATTTG-3′; Reverse, 5′-GAAACCAGGGAGGTCTGATTC-3′.

**Proteomic analysis**. Testicular tissue for proteomic analysis was obtained from 8-week-old C57BL/6 male mice and subjected to LC-MS/MS analysis with a standard protocol. In brief: (1) The mouse testes were ground using liquid nitrogen and transferred to tubes containing lysis buffer (8 M urea, 1% Protease Inhibitor Cocktail) on ice, followed by sonication using a high intensity ultrasonic processor (Scientz). The supernatant was collected after centrifugation, and the protein concentration was determined; (2) After trypsin digestion and purification, peptides were reconstituted in 0.5 M TEAB and processed according to the manufacturer's protocol using the TMT kit; (3) Released peptides were subjected to LC-MS/MS identification and quantification on a Thermo Scientific LTQ Orbitrap XL mass spectrometer (Thermo Fisher Scientific) with a Finnigan Nanospray II electrospray ionization source; (4) The raw data were processed using Domain Annotation (InterProScan), GO Annotation (http://www.ebi.ac.uk/GOA/), and KEGG Pathway Annotation (KEGG online service tools KAAS mapper).

**Phosphoproteomics analysis**. Testicular tissue for phosphoproteomics analysis was obtained from 8-week-old C57BL/6 male mice and subjected to LC-MS/MS analysis with a standard protocol. In brief: (1) The mouse testes were ground using liquid nitrogen and transferred to tubes containing lysis buffer (8 M urea, 1%

Protease Inhibitor Cocktail) on ice, followed by sonication using a high intensity ultrasonic processor (Scientz). The supernatant was collected after centrifugation, and the protein concentration was determined; (2) After trypsin digestion and purification, peptides were reconstituted in 0.5 M TEAB and processed according to the manufacturer's protocol using the TMT kit; (3) The peptides were first separated into 60 fractions by a gradient of 8–32% acetonitrile (pH 9.0) for 60 min and then pooled into four fractions. The IMAC microspheres with enriched phosphopeptides were then collected by centrifugation. After eluting the enriched phosphopeptides by vibration with elution buffer containing 10% $NH_4OH$, the supernatant containing phosphopeptides was collected and lyophilized for LC-MS/MS analysis; (4) Released peptides were subjected to LC-MS/MS identification and quantification on a Q Exactive HF-X mass spectrometer (Thermo Fisher Scientific) with an NSI source; (5) The raw data were processed using Domain Annotation (InterProScan), GO Annotation (http://www.ebi.ac.uk/GOA/), and KEGG Pathway Annotation (KEGG online service tools KAAS mapper).

**RNA-sequencing analysis**. The testis samples from 8-week-old C57BL/6 male mice were washed in 1×phosphate-buffered saline (PBS). Total RNAs were extracted from the samples using the Dynabeads mRNA Purification Kit (61006, Thermo Fisher Scientific). Ribo-Zero™ rRNA Removal Kit (MRZPL1224, llumina) was used to remove rRNA of samples. RNA integrity was evaluated using the Agilent 2100 Bioanalyzer system (Agilent Technologies). An mRNA sequencing library for RNA-seq was constructed using the TruSeq RNA Library Prep kit v2 (RS-122-2001, Illumina), followed by paired-end (2 × 100 bp) sequencing using the Illumina HiSeq 4000 Sequencing System (Illumina) in the Beijing Genomics Institute. Raw paired-end reads were filtered by FASTX-Toolkit. The quality of the reads was confirmed by FastQC (ver:0.11.3). Raw data (raw reads) of fastq format were processed through in-house perl scripts. Feature Counts (ver: l.5.0-p3) was used to count the reads numbers mapped to each gene. FPKM of each gene was calculated based on the length of the gene and reads count mapped to this gene. Signaling pathway matching analysis for the differentially expressed gene (DEG) list was performed using KEGG Mapper in Kyoto Encyclopedia of Genes and Genomes (KEGG, https://www.genome.jp/kegg/). The mapping of GO to DEGs was carried out using Blast2GO (ver: 4.1.9).

**Generation of animal models**. The animal experiments were approved by the Experimental Animal Management and Ethics Committee of West China Second University Hospital, Sichuan University. All animal procedures complied with the Animal Care and Use Committee of Sichuan University. Mouse housing is in an environment with a room temperature range between 20 and 26 °C under a 12-h light: 12-h dark cycle, and a relative ambient humidity at the level of mouse cages of 40–70%. One-month C57BL/6 female mice were superovulated and mated with 8–10 weeks old C57BL/6 male mice to collect zygotes. Cas9 was amplified from pX260 (Addgene) using KOD-Plus-Neo (KOD-401, TOYOBO), purified with Universal DNA Purification Kit (DP214, TIANGEN) and transcribed in vitro using mMESSAGE mMACHINE T7 ULTRA kit (AM1345, Life Technologies). sgRNA with T7 promoter was amplified from pX330 (Addgene) using KOD-Plus-Neo, purified with the Universal DNA Purification Kit and transcribed in vitro using the MEGAshortscript T7 kit (AM1354, Life Technologies). After transcription, Cas9 and sgRNA were purified with the MEGAclear kit (AM1908, Life Technologies) according to the manufacturer's instructions. *Cep128* mutant mice were generated by injecting Cas9, sgRNA and donor template into zygotes. After injection, the zygotes were cultured in KSOM (MR-106-D, Millipore) at 37 °C under 5% $CO_2$ to reach the 2-cell stage, followed by embryo transfer into oviducts of pseudopregnant ICR females at 0.5-day postcoitum (dpc). *Cep128* mice with point variant or frameshift variant were selected and maintained. Detailed information regarding primers for generation of animal models is shown in Supplementary Table 5.

**RNA isolation and quantitative PCR**. Total RNA was isolated using the TRIzol reagent (15596026, Thermo Fisher Scientific) and converted to cDNA with SuperScript™ IV Reverse Transcriptase (18090010, Thermo Fisher Scientific) according to the manufacturer's instructions. qPCR was performed using KiCq-Start SYBR Green qPCR ReadyMix (KCQS00, Sigma-Aldrich) on an iCycler RT-PCR Detection System (Bio-Rad Laboratories). qPCR data were normalized using the $2^{-\Delta\Delta Ct}$ method. Primer sequences are indicated in Supplementary Table 6.

**Cell culture and plasmid construction**. HEK293T cells and hRPE-1 cells were obtained from the American Type Culture Collection (ATCC® CRL-11268™; ATCC® CRL-4000™). HEK293T cells were grown in DMEM (11965092, Gibco) supplemented with 10% fetal bovine serum (FBS) (F8318, Sigma-Aldrich). hRPE-1 cells were grown in DMEM/F12 (11320033, Thermo Fisher Scientific) supplemented with 10% FBS, which were cultured in serum-free medium for 48 h to induce cilium formation. The expression plasmids encoding WT *CEP128* (NM_152446.5) (pENTER-Flag-WT-*CEP128*) were constructed by Vigene Biosciences (Jinan, China). The variant (c.665G > A, p.R222Q) was introduced into WT-*CEP128* plasmid using the site-directed KOD-Plus-Mutagenesis Kit (SMK-101, Toyobo) according to the manufacturer's instructions. The plasmids were transfected with Lipofectamine 3000 (L3000015, Invitrogen) for cell lines according to the manufacturer's protocol.

**Western blot and co-immunoprecipitation**. Cells and mouse testes tissues were lysed in RIPA buffer (P0013B, Beyotime) supplemented with Halt™ Protease Inhibitor Cocktail (78425, Thermo Fisher Scientific) and phosphatase inhibitor cocktail (P5726, Sigma-Aldrich). Lysates were mixed with SDS Sample loading buffer (P0015, Beyotime) and boiled for 10 min. Following SDS-PAGE separation, resolved proteins were transferred to PVDF membrane (IPVH00010, Millipore). The membrane was blocked for 30 min at room temperature and incubated consistently with primary and secondary antibodies diluted in 5% milk 1×TBS-T. Finally, darkroom development techniques for chemiluminescence were used for acquiring image.

For co-immunoprecipitation, the extracted proteins were generated using methods similar to the western blot assays. Lysates were incubated with target antibodies on a rotating wheel overnight at 4 °C. Next, Protein A/G magnetic beads (88802, Thermo Fisher Scientific) were added to each incubation sample for 1 h at room temperature. The beads were washed three times with wash buffer (20 Mm HEPES, 15% glycerol, 250 mM KCl, 0.2 mM EDTA, 1% NP-40). The co-immunoprecipitated proteins were eluted by 1 × SDS sample and analyzed by immunoblot as indicated.

**Isolation of mouse spermatogenic cells**. Spermatogenic cells were obtained through cell diameter/density at unit gravity using the STA-PUT velocity sedimentation method as previously described[76,77]. In brief, mouse germ cells were extracted from the seminiferous tubules of adult C57BL/6 male mice. Then, spermatogenic cells were resuspended in 25 ml of 0.5% BSA solution and filtered through an 80 mm mesh to remove cell aggregates. After passage through mesh filter, the cells were resuspended in buffer containing 0.5% BSA and loaded in a STA-PUT velocity sedimentation cell separator (ProScience) for gradient separation. Several germ cell populations were collected for subsequent analysis.

**Immunofluorescence staining**. For cell staining, the samples were washed in 1 × PBS (3 × 10 min), blocked in PBSTB (0.1% Triton X-100, 3% bovine serum albumin in 1 × PBS) for 1 h at room temperature, and incubated with primary antibodies overnight at 4 °C. Washes were performed using 1 × PBS (3 × 10 min); then, the samples were incubated with Alexa Fluor 488 (1:1000, A21206, Thermo Fisher Scientific) or Alexa Fluor 594 (1:1000, A11005, Thermo Fisher Scientific) labeled secondary antibodies for 1 h at room temperature, and stained with 4,6-diamidino-2-phenylindole (DAPI, 28718-90-3, Sigma-Aldrich) for 10 min. Images were captured with a confocal microscope (Olympus FV3000).

For spermatozoa staining, the samples were coated on the slides covered with poly-lysine (ST509, Beyotime) and Sigmacote coverslips (SL2, Sigma-Aldrich) were placed on top. The slide was quickly frozen and stored in liquid nitrogen for 10 min. Then the coverslip was carefully removed with tweezers and the slide was immersed in pre-cooled methanol for 2 min. After washing three times with 1 × PBS, the slide was placed in fresh 1 × PBS with 3% Triton X-1000 for 60 min at room temperature, then permeabilized with 3% bovine serum albumin for 30 min at room temperature. Next, the samples were incubated with primary antibodies overnight at 4 °C, followed by 1 h incubation at 37 °C with secondary antibodies and 0.5% DAPI. Images were captured with a confocal microscope (Olympus FV3000).

For staining of testicular tissues, samples were fixed in 3.7% buffered formaldehyde. After fixation, the tissues were dehydrated through an ethanol gradient and embedded in paraffin. The samples were sectioned at a thickness of 5 μm. After being deparaffinised and rehydrated, sections were treated with 3% hydrogen peroxide for 10 min at room temperature and with 20 mM sodium citrate for 15 min at 95 °C, followed by overnight cooling. Primary antibodies were applied at suitable dilutions at 4 °C overnight, followed by incubation with biotinylated secondary antibodies for 1 h and DAPI for 30 min. Images were captured with a confocal microscope (Olympus FV3000).

**Histological examination**. Testicular, epididymal and ovarian tissue from 8-week-old C57BL/6 male and female mice were fixed with 4% paraformaldehyde for 8 h. After dehydration by ethanol, they were embedded in paraffin and sectioned at 5 μm. The sections were stained with haematoxylin and eosin and observed under a microscope (Zeiss, Axio Imager 2).

**Antibodies**. Details of the specific antibodies that were used in the study are as follows: anti-CEP128 (HPA001116, Sigma-Aldrich, WB: 1:1000, IF: 1:200); anti-Ac-Tubulin (ab24610, abcam, IF: 1:1000); anti-NIN (A8215, ABclonal, WB: 1:1000); anti-CEP170 (27325-1-AP, Proteintch, WB: 1:500); anti-YY1 (sc-7341, Santa Cruz Biotechnology, WB: 1:500); anti-GR-β (sc-393232, Santa Cruz Biotechnology, WB: 1:500); anti-SEPT4 (Septin 4) (A10238, ABclonal, IF: 1:100); anti-RCBTB2 (13225-1-AP, Proteintech, WB: 1:1000, IF: 1:50); anti-WBP2NL (22587-1-AP, Proteintech, WB: 1:500, IF: 1:50); anti-CRISP1(MAB4675, R&D Systems, WB: 1:500, IF: 1:50); anti-PRSS55 (bs-19443R, Bioss, WB: 1:1000, IF: 1:50); anti-ubiquitin (ab7780, Abcam, WB: 1:500), anti-GAPDH (ab8245, Abcam, WB: 1:1000); anti-CEBPβ (sc-7962, Santa Cruz Biotechnology, WB: 1:500); anti-Centriolin (sc-365521, Santa Cruz Biotechnology, WB: 1:500).

**Electron microscopy**. For SEM, samples were fixed with 2.5% glutaraldehyde in 0.1 M sodium cacodylate (pH 7.2) for 1 h, then dehydrated using progressive ethanol concentrations (35, 50, 75, 90, 95, and 100%) for 10 min each and dried using a $CO_2$ critical-point dryer (Eiko HCP-2, Hitachi). Finally, the samples were observed under the SEM (Hitachi S3400) at an accelerating voltage of 15 kV.

For TEM, samples were fixed in 3% glutaraldehyde, phosphate-buffered to pH 7.4 and post-fixed with 1% $OsO_4$. After dehydration, the samples were incubated in propylene oxide followed by embedding in a mixture of Epon 812 and Araldite. Ultrathin sections obtained by an Em UC6 Ultramicrotome (Leica) were collected on TEM nickel grids and analyzed using a TEM (TECNAI G2 F20, Philips) at 120 kV.

**Intracytoplasmic sperm injection (ICSI)**. Mice were housed in a temperature-controlled environment under a 12-h light: 12-h dark cycle. The animal experiments were approved by the Experimental Animal Management and Ethics Committee of West China Second University Hospital, Sichuan University. All animal procedures complied with the procedures of the Animal Care and Use Committee of Sichuan University. One-month female KM mice were super-ovulated by using 5 IU equine CG (HOR-272, ProSpec) followed 48 h later with 5 IU human CG (hCG) (M2530, Easycheck). Metaphase II-arrested (MII) oocytes were collected 13–14 h after hCG administration and incubated until use in Chatot-Ziomek-Bavister medium (M2750, Easycheck) at 37.5 °C in an atmosphere of 5% $CO_2$ in air.

Cauda epididymal sperm were collected from males into HTF medium (M1150, Easycheck). Sperm tails were removed by repeated freezing and thawing. To perform ICSI, a single sperm head was microinjected into an MII oocyte by using a NIKON inverted microscope (Tokyo, Japan) and a Piezo (PrimeTech, Osaka, Japan). The injection was performed in Whitten's-Hepes medium containing 0.01% polyvinyl alcohol (Whitten's-Hepes-PVA) (12360-038, Gibco), with 3.5 μg/ml cytochalasin B (C-6762, Sigma-Aldrich). After injection, the oocytes were moved into the G1-Plus medium (10132, Vitrolife) at 37.5 °C in an atmosphere of 5% $CO_2$ in air.

**Statistical analysis**. Statistical analyses were conducted using GraphPad Prism 8.4.0 software and SPSS 17.0 software. All data were presented as the mean ± SEM. A $p$ value of <0.05 was considered statistically significant. Statistical significance between two groups was calculated using an unpaired, parametric, two-sided student's $t$ test.

**Reporting summary**. Further information on research design is available in the Nature Research Reporting Summary linked to this article.

## Data availability

All relevant data that support the findings of this study are available in a publicly accessible repository. The proteomic analysis data generated in this study have been deposited in the Proteomics Identifications Database under accession code 'PXD022731'. The phosphoproteomics data used in this study are available in the Proteomics Identifications Database database under accession code 'PXD025330'. The RNA-seq data generated in this study have been deposited in the SRA Database under accession code 'PRJNA799590'. The whole exon sequencing data generated in this study have been deposited in the SRA Database under accession code 'PRJNA719582'. Web links for publicly available datasets: ExAC Browser (http://exac.broadinstitute.org), 1000 Genomes Project (https://www.internationalgenome.org/), gnomAD databases (http://gnomad-sg.org/), dbSNP (https://www.ncbi.nlm.nih.gov/snp/), KEGG (https://www.genome.jp/kegg/), Gene Ontology (GO) (http://geneontology.org/), HGMD (http://www.hgmd.cf.ac.uk/ac/all.php). Source data are provided with this paper.

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

## Acknowledgements

We thank the patients, their family members, and the control group for their support during this research study. We thank the Analytical and Testing Center of Sichuan University for the morphology characterization and the authors would be grateful to Guiping Yuan for her help of TEM images. This study was funded by the National Natural Science Foundation of China (31701085 and 31625015).

## Author contributions

Y.S. designed and supervised the study experiments. Y.Y., X.J., Y.W., H.L., L.Q. and Jinghong L. collected data and conducted the clinical evaluations. Xueguang Z., Y.M. and M.L. performed experiments and analyzed most of the data. L.W., Xiaojian Z. and X.S. generated the CRISPR mice. C.J., Y.Liu, Jinsong L., Y.Li, R.Z., Yongkang Sun., J.S., X.L. and Y.Liang performed experiments. Y.S. wrote the manuscript, with input from others. F.Z. provided valuable advice for the manuscript. All authors revised and approved the article.

## Competing interests

The authors declare no competing interests.
