## [Peer Review File · Nature Communications]

CEP128 is involved in spermatogenesis in humans and miceREVIEWER COMMENTS

Reviewer #1 (Remarks to the Author):

Dysfunction of CEP128 impairs spermatogenesis but not ciliogenesis in human and mice

1. This is an interesting article on the identification of homozygosity for a missense variant in two siblings that with defective spermatogenesis. The authors then modeled loss of function and homozygosity for the variant in mouse models that also had defective spermatogenesis. The strengths in the article that include very clear and well-done images of sperm morphology by light microscopy and electron microscopy.
2. The reviewer has difficulty with the title of manuscript. The authors show in Figure 3b that over expression of the mutant protein in RPE cells resulted in short or no cilia. The authors note that previous reports suggest that over expression of CEP128 could suppress ciliation.
3. Consider addressing the abstract. The abstract does not clearly inform the readers about what the authors found. For example, in one family homozygosity for variant in CEP128 associated with abnormal sperm, variant lead to increased stable protein, etc.
4. Same issue in abstract regarding conflict on whether this variant affect ciliogenesis.
5. Page 4, introduction. First sentence is confusing and should be rewritten. Basal bodies are mother and daughter centrioles.
6. Page 4, lines 68-70. There are multiple very well written reviews on the role of CEPs and the role of CEPS extend well beyond cancer and microcephaly. This needs to be reworked.
7. Page 5, line 86. The R222Q change should be referred to as a variant throughout the manuscript.
8. Page 5, lines 95-97, it is not clear that the authors proved that the CEP128 regulate genes associated with fertility. Modify sentence to address precisely what genes had decreased transcriptional levels.
9. Page 6. MutationTaster uses a statistic which should be included. When the reviewer placed the variant in MutationTaster is was predicted to be a polymorphisim. SIFT did predict damaging. Other databases predicted tolerated – one of the limitations of all prediction programs is some degree of inconsistency.
10. Page 6. The reviewer is raising concerns regarding paragraph on the bottom of page 6 to page 7. The variant is not absent from human databases. Its highest allele count is in east asians with an allele frequency of 0.00893.
11. Page 7 – agree that the CEP128 signal is different in Figure 2D but immunofluorescence is not quantitative – it is diffuse.

12. Page 7 and Figure 3a. While it does look like there is more mutant protein when compared to control, this is not quantified. While in the check list the authors note that replicates and statistics were performed, this is not included in the results in a meaningful way.
13. Page 8, Figure 3b. What there is no control of overexpression of WT CEP128.
14. Page 8, Figure 3b. What do the authors suggest is abnormal nuclear morphology?
15. Page 8, Figure 3c. Why did the authors suspect decreased expression of NIN, centrin, CEP170? The figure is of poor quality, not convincing for decreased expression, particularly for CEP170 and again, no evidence for technical replicates and no statistics.
16. Page 8, lines 167-169 (Figure 3d) The levels of decreased RNA levels do not correlate with the western blot analysis by visualization.
17. Page 8-9, lines 169-174, Figure 3e. Similar concerns, no evidence that there are decreases in any of the proteins with no statistics or evidence of replicates.
18. Page 9, Sup Fig 1a. Appreciate the western blot and agree with the interpretation, but testes is known to highly express proteins in general.
19. Page 11, lines 226-229. What are respiratory organs? What do the authors mean by no obvious ciliary defects?
20. Page 13, Figure 5 – No statistics, similar concern.
21. Page 16 – concern re: genetics for the heterozygous variants are not consistent with the findings that the homozygous variant causes infertility. 500 controls are not adequate for rare variants that may or may not be pathogenic.
22. Page 17, lines 350-351. What key molecules are being regulated?
23. Page 17, line 358-359. Authors note that high levels of CEP128 hampered cilia growth which is not consistent with the title and thoughts through the paper.
24. Page 17, lines 360-361. The authors do not show evidence for cilia length or number in their figures and no statistics have been performed.
25. Page 18. Ciliopathies are complex disorders that affect different organ systems differently and do not necessarily correlate with level of expression since almost all vertebrate cells have cilia. For example, Suppl Figure 1 shows higher levels of expression in heart and the eye, yet the patients have not phenotype.
26. It would be helpful if in the introduction and discussion that spermatogenesis was explained so the reader who is not an expert in male infertility could better understand.
27. Usually there are excel data tables included of all the proteomic findings (TMT).
28. Did the authors look at TGF beta/BMP signaling since CEP128 has been shown to be involved (PMID 29514088).

Minor Suggestions

1. Abstract – remove “could” because it is not definitive.
2. Introduction – page 4. Centrosome has multiple functions.
3. Page 5, lines 91-93 are redundant with the previous sentences.
4. Page 6, line 121, remove “remarkably”.
5. Page 7 – line 128, change to confirmed homozygosity for the CEP128 variant. The parents are heterozygous for the variant.
6. Page 7, line 141, remove “strikingly?”
7. Page 10, line 200, change “irrespective” to “does not play a role”
8. Page 11, line 233, change obtaining to “to generate”

Reviewer #2 (Remarks to the Author):

In this study, a homozygous missense mutation of the CEP128 gene was identified in two siblings with primary male infertility related to cryptozoospermia from a consanguineous family through whole-exome sequencing. The mutation was then studied in two systems with three approaches: overexpression in tissue culture cells and KI in mice, and KO in mice. In tissue culture cells, the mutant CEP128 appears overexpressed, forms aggregates, abnormal cilia, nuclei, and possibly affects ceps and transcription levels. All this suggests the mutant protein has a gain of function mutation that has a toxic effect on the NC cells. Mouse model harboring the orthologous missense variant via CRISPR-Cas9 gene editing, knock-in (KI) mice, exhibited reduced sperm counts and spermatozoa with morphological abnormalities, and male infertility. Analysis of the KO CEP128 finds a milder phenotype than the KI, suggesting that the function phenotype's gain is more severe than that of the LOF phenotype. Analysis of zygote produced by ICSI fertilization with the KI and KO mutant shows that the abnormal sperm cannot support embryo development. The paper provides convincing evidence that the missense mutation of the CEP128 is a pathological mutation. The mutation is a gain of function, and it has a phenotype similar in its specific effect on sperm but different from the loss of role in several cell biological details. Potentially the particular impact is because CEP128 is highly enriched in the testis compared to other cell types.

The paper shows that Cep128 plays a crucial role in spermatogenesis but is not essential for viability or cilia in general. The reasoning for the tissue difference is unclear, considering that CEP128 is thought to be a general centriole appendage protein. Also, the role of CEP128 in spermatogenesis is unknown, and

it is not clear if it purely due to its centriolar function. Why the overexpression of WT-CEP128 or mutant CEP128 leads to the altered expression and toxic phenotype is unclear? Why the KI and KO mutant fertilized embryo failed to develop is unknown?

The paper has many interesting findings that can be further developed but appear to have a critical mass of information that is likely to make it attractive to Nature Communication's audience once all the minor points below are addressed.

Minor points:

- Unconventional description/statements:

“centrosomal protein (CEP) family”- centrosomal protein (CEP) are not family of protein – they are functional or structure associated group.

“33 CEPs have been identified” – this need reference and detection as many will claim that there are hundreds of centrosomal proteins.

“few are related to microcephaly4.”- I would say surprisingly, several of them are related to microcephaly.

“dysplasia of the fibrous sheath (DFS) and globozoospermia is always accompanied by centrosome dysfunction7” - I would say in many – not always

- There is a need to provide information on the role of Cep128 in the centriole as a subdistal appendage.

“Initially, no spermatozoa were observed in the replicate wet preparations” – what is wet preparations? Is it a semen sample?

- The paper's main text should start with Fig 1 and not Fig 2.

- Fig. 1 – please define in the legend all the elements in the figure, including III-3 and III-2

- Fig. 2c – it is essential to have similar picture orientations in the normal control and mutants – without that, it is hard to assess the sperm connecting piece phenotype. Also - reorient the individual panels, so they all have the same orientation for each subcategory. Mark the location of PC and DC in each panel and describe their phenotype. Also, label the mitochondria.

- Fig 2b “The site of the CEP128 missense mutation is evolutionally conserved among various species” – please explain if they are all mammals – what happened to the amino acid in more distinct evolutionary groups like insects and worms?

- “the spermatozoa from the two patients represented the lack of the central-pair microtubules (CPs)”. One of the patients has clear central-pair microtubules – please correct the sentence.

- Fig. 2d – is the CEP128 localize to the distal or proximal centriole?

- “With the accumulation and ectopic expression of EGFP-CEP128 mutant in the cytoplasm short cilia” show examples of short cilia – this phenotype needs quantification and statistical analysis.
- 1a-c and Fig 2 a, b, c, and many other figures and descriptions (such as nuclear morphology) need quantification and statistical analysis.
- Fig 3b needs insets with zoom on basal body and cilia
- Define Pc
- What is “sperm ultrastructural disarray”?
- The paper legends are poorly written and need extensive work.
- “ICSI failed in these patients because of the aberrant proximal centrioles exhibited in almost all the spermatozoa” – add an explanation on the possible effect on distal atypical centriole.

Tomer Avidor-Reiss

Reviewer #3 (Remarks to the Author):

The manuscript by Ying Shen and colleagues presents convincing evidence for a role of CEP128 in human and mouse spermatogenesis, and for a mutant (R222Q) probably causing male infertility in human. Overall, the manuscript is well written and the figures are very nice. The results are novel and potentially important in terms of male infertility diagnosis and treatment. The mechanistic part of the study is less convincing. It lacks important methodological details and intermediate results, and the logic is sometimes difficult to follow.

Please find below a few suggestions for improvement:

In contrast to what is mentioned on Page 6 lines 123-126 (“This mutation is absent in all of the human population genome databases”), this mutation is already described in GnomAD (<https://gnomad.broadinstitute.org/variant/14-81329198-C-T>) and dbSNP (rs114668642). Please revise the text accordingly.

The title “Dysfunction of CEP128 impairs spermatogenesis but not ciliogenesis” is a bit confusing: Fig 3 confirms that CEP128 is involved in the formation of cilia in RPE cell lines. In addition, the flagellar

defects observed in CEP128 KO/KI mice also suggest that CEP128 is involved in flagella biogenesis. The presented results suggest that CEP128 dysfunction does not impair ciliogenesis in respiratory organs, but do not exclude effects on ciliogenesis in other structures.

The expression of CEP128 is clearly downregulated (but not absent; this should be stressed in the text) in testes of KO mice and upregulated in testes of KI mice (qPCR and WB). CEP128 expression should also be checked in oocytes and lung of KO and KI mice before concluding that CEP128 does not play any role in these tissues (page 10 line 200, page 11 line 227).

A large part of the presented results relies on immunofluorescence analysis of human and mice samples performed with HPA001116 antibody. This antibody has already been used in human cell lines, but I could not find any validation study in human sperm, or in any mouse sample. The authors should provide at least a full-range western blot showing a single band at the expected MW in human sperm and in sperm from WT, KO and HI mice. They should also show that HPA001116 reproduces the results obtained with the Flag antibody on Figure 3a. Page 7 line 132: please precise what “markedly enhanced” means. Could you provide any quantitative measurement of the signal?

Please provide details about the CEP128 construct used for cell line transfection (either the full sequence or the database accession number of “wild type CEP128)

The list of the 399 proteins shown to be differentially expressed by the proteomics approach should be provided, with fold changes. How many proteins were up/down in KO and KI mice? What fold change and p value were considered for a differential expression? The choice of the five proteins to follow up appears quite subjective. Were those proteins the ones with the highest fold change? Why not choosing all the ones presented in the Reproduction diagrams Fig 7b/c ? Or the ones also found with the RNA analysis?

The methodological details about the RNA sequencing of mouse testes and the differential expression analysis are missing. Please provide the full lists of the differentially regulated genes, with fold changes. How do these lists compare with the lists obtained with the proteomics approach? How do the five candidates selected above behave at transcript level?

The Co-IP results are interesting but rather preliminary. A putative link between the formation of the complex and ubiquitination-mediated degradation of the members of the complex is missing. Is one of the complex members known to be involved in ubiquitination-dependent pathways? Did you identify proteins involved in ubiquitination pathways in your differential expression analyses?

A recent study by Mönnich et al. (29514088) suggested that CEP128 affects TGF- β 1-induced phosphorylation of multiple proteins that regulate cilium-associated vesicle trafficking. Would this be compatible with your results? A quantitative phosphoproteomics analysis of WT/KO/KI mouse testes might help deciphering the molecular mechanisms of CEP128 function in spermatogenesis.

Reviewer 1's comments:

Q1: This is an interesting article on the identification of homozygosity for a missense variant in two siblings that with defective spermatogenesis. The authors then modeled loss of function and homozygosity for the variant in mouse models that also had defective spermatogenesis. The strengths in the article that include very clear and well-done images of sperm morphology by light microscopy and electron microscopy.

A1: We thank the reviewer to approve our work!

Q2: The reviewer has difficulty with the title of manuscript. The authors show in Figure 3b that over expression of the mutant protein in RPE cells resulted in short or no cilia. The authors note that previous reports suggest that over expression of CEP128 could suppress ciliation.

A2: Thanks to the reviewer's comment! Our observation that over-expression of the mutant protein in RPE cells resulted in short or no cilia confirmed the deleteriousness of this variant *in vitro*. *In vivo*, we found that deleterious *CEP128* variants could impair spermatogenesis in humans and mice, but the cilia of lung, trachea, brain and eye from mouse models were normal and the patients carrying homozygous *CEP128* p.R222Q variant presented that they had no other phenotypes except infertility. Furthermore, a previous study showed that another centrosomal protein CEP131 (which is a different protein from CEP128 in this study) localizes to centriolar satellites, and loss of CEP131 impairs ciliogenesis in mouse fibroblasts *in vitro*. However, cilia throughout *Cep131* null mice are functionally normal, as embryonic patterning and adult homeostasis are grossly unaffected. Surprisingly, the *Cep131* null mice exhibit the abnormal flagellogenesis, resulting in male infertility (Hall, E. A. et al. Acute versus chronic loss of mammalian Azi1/Cep131 results in distinct ciliary phenotypes. *PLoS Genet* 2013; 9, e1003928). Therefore, we suggested that *CEP128* p.R222Q variant perturbing cilia growth in cultured cells but not the cilia development of other organs (except the testis) *in vivo*, because the regulation mechanism of cilia growth *in vivo* is more complex and other centrosomal proteins but not CEP128 may play the key roles in ciliogenesis in these organs, or a

compensation mechanism exists to allow ciliogenesis to proceed despite the lack of CEP128. The manuscript has been revised accordingly.

In the previous title, we said that “Dysfunction of CEP128 impairs spermatogenesis but not ciliogenesis in humans and mice”. This title is a bit confusing, because the flagellum is a special cilium. Therefore, we have also modified the title in the revised manuscript.

Q3: Consider addressing the abstract. The abstract does not clearly inform the readers about what the authors found. For example, in one family homozygosity for variant in CEP128 associated with abnormal sperm, variant lead to increased stable protein, etc.

A3: We thank the reviewer for the expert comments! We have rewritten the abstract in the revised manuscript.

Q4: Same issue in abstract regarding conflict on whether this variant affect ciliogenesis.

A4: We are sorry for bringing the doubts. *In vitro*, over expression of the mutant CEP128 protein resulted in short or no cilia in RPE cells. *In vivo*, the flagellar defects observed in *Cep128* KI mice suggest that this variant is involved in flagellar development, while the distribution, morphology and ultrastructure of cilia in other organs, such as lung, trachea, eye, and brain, were normal. Therefore, this variant only affects ciliogenesis (flagellogenesis) in the testes *in vivo*. We have rewritten the abstract in the revised manuscript accordingly.

Q5: Page 4, introduction. First sentence is confusing and should be rewritten. Basal bodies are mother and daughter centrioles.

A5: We thank the reviewer for expert suggestions! We have rewritten the introduction part in the revised manuscript.

Q6: Page 4, lines 68-70. There are multiple very well written reviews on the role of CEPs and the role of CEPS extend well beyond cancer and microcephaly. This needs to be reworked.

A6: We thank the reviewer for expert suggestions! We have added more diseases resulting from dysfunction of centrosomal proteins into the introduction part of the revised manuscript.

Q7: Page 5, line 86. The R222Q change should be referred to as a variant throughout the manuscript.

A7: We thank the reviewer for expert suggestions! We have changed “mutation” into “variant” in the revised manuscript.

Q8: Page 5, lines 95-97, it is not clear that the authors proved that the CEP128 regulate genes associated with fertility. Modify sentence to address precisely what genes had decreased transcriptional levels.

A8: We thank the reviewer for expert comments! In the previous study, we applied both proteomics and RNA-seq assays to investigate the key molecules that might contribute to male infertility in *Cep128* KO and KI mice. We found the significantly decreased expression of *Wbp2nl*, *Rcbtb2*, *Prss55*, *Crisp1* and *Defb22* by proteomic assay. Furthermore, RNA-seq revealed the diminished transcriptional levels of *Cst8*, *Wnt3*, *Lrrc52*, *Eqtn*, *Calr3*, *Sox30*, *Tnp2*, *Sun5*, *Irs4*, *Catsper1*, *Lmnb2*, *Spata25*, *Upk3b*, *Spata31*, *Ccdc192*, *Ccdc63*, *Tssk4*, *Cdo1*, *Inpp4b*, *Spata48*, *Tmem119*, *Odf3*, *Spata9*, *Tmem95*, *Insl5*, *Klc3* and *Defb22*. These genes are involved in human and/or mouse spermatogenesis, including sperm flagellar development and sperm production as well as in fertilization.

According to the reviewers' expert suggestion, we have performed quantitative phosphoproteomics analysis on *Cep128* KO/KI/WT mouse testes to further decipher the potential molecular mechanisms underlying the CEP128 function in spermatogenesis since CEP128 loss has been suggested to decrease TGF- β /BMP-induced phosphorylation of multiple proteins that regulate cilium-associated vesicle trafficking. Importantly, we found TGF- β /BMP signaling members (including *Lefty2*, *Neo1*, *Fbn1*, *Rbl1*, *Gata4* and *Trim33*) exhibited greatly

reduced phosphorylation in KI mice compared to WT mice. Rbl1, Gata4 and Trim33 have been reported to be involved in the regulation of spermatogenesis. According to the reviewers' expert suggestion, we have listed some of these key molecules and added more discussions in the revised manuscript.

Q9: Page 6. MutationTaster uses a statistic which should be included. When the reviewer placed the variant in MutationTaster it was predicted to be a polymorphism. SIFT did predict damaging. Other databases predicted tolerated – one of the limitations of all prediction programs is some degree of inconsistency.

A9: We thank the reviewer for expert comments! The prediction results of Mutation Taster, SIFT, PolyPhen-2, and CADD have been added in Supplementary Table 1 in the revised manuscript. SIFT, PolyPhen-2, and CADD predict damaging of this variant. The functional prediction results of this variant are variable between different *in silico* tools. This inconsistency may reflect the different evidence and algorithms employed in these tools.

Instead, this *CEP128* variant is absent or very rare in human populations (e.g., 0.00051% in the gnomAD database) and only the affected individuals in the family carry the homozygotes (no homozygote of this variant was identified in the gnomAD database). Moreover, the *in vitro* experiment confirmed the deleteriousness of the variant in ciliogenesis. Most importantly, our *Cep128* KI mice did show the infertile phenotypes. Therefore, our *in vitro* and *in vivo* functional assays did suggest that this *CEP128* variant is pathogenic.

More data and discussions have been added to the revised manuscript.

Q10: Page 6. The reviewer is raising concerns regarding paragraph on the bottom of page 6 to page 7. The variant is not absent from human databases. Its highest allele count is in east asians with an allele frequency of 0.00893.

A10: We thank the reviewer for kind reminder of the allele frequency in East Asians. The updated data showed that this variant is absent in African, Europe, South Asian and American

populations and at a low frequency in Asians in ExAC Browser, 1000Genomes and gnomAD databases. We have added these updated data to in supplementary Table 1 in the revised manuscript.

Q11: Page 7 – agree that the CEP128 signal is different in Figure 2D but immunofluorescence is not quantitative – it is diffuse.

A11: We thank for the reviewer's expert comments! We have supplemented the statistics of fluorescence intensity of Figure 2d by ImageJ in the revised manuscript. Due to the extremely low sperm counts of the patients, we could not investigate the CEP128 expression of the sperm by western blot.

Q12: Page 7 and Figure 3a. While it does look like there is more mutant protein when compared to control, this is not quantified. While in the check list the authors note that replicates and statistics were performed, this is not included in the results in a meaningful way.

A12: Thank for the reviewer's suggestion! We have quantified the results of western blot in Figure 3a in the revised manuscript.

Q13: Page 8, Figure 3b. What there is no control of overexpression of WT CEP128.

A13: According to the reviewer's suggestion, the control of overexpression of WT-CEP128 has been added in Figure 3b in the revised manuscript.

Q14: Page 8, Figure 3b. What do the authors suggest is abnormal nuclear morphology?

A14: When we overexpressed mutant CEP128 in RPE1 cells, we observed that most of the cells showed nuclear pyknosis, even some represented nuclear fragmentation. Therefore, we

suggested abnormal nuclear morphology. We have added this description in the revised manuscript.

Q15: Page 8, Figure 3c. Why did the authors suspect decreased expression of NIN, centrin, CEP170? The figure is of poor quality, not convincing for decreased expression, particularly for CEP170 and again, no evidence for technical replicates and no statistics.

A15: According to the reviewer's suggestion, we have performed western blot again, got the high-quality images, and added the statistics (Figure 3c) in the revised manuscript. Here, we also provided the evidence for technical replicates (Fig R1).

Figure R1. Diminished levels of NIN, centriolin, and CEP170 were detected in the cells transfected with WT-*CEP128* (WT) or *CEP128*^{R222Q} (Mut) when compared to the negative control (NC) by western blot analysis. Three replicates were shown here.

Q16: Page 8, lines 167-169 (Figure 3d) The levels of decreased RNA levels do not correlate with the western blot analysis by visualization.

A16: Thanks to the reviewer's suggestion! We have performed the qPCR again and showed the reliable results in Figure 3d in the revised manuscript.

Q17: Page 8-9, lines 169-174, Figure 3e. Similar concerns, no evidence that there are decreases in any of the proteins with no statistics or evidence of replicates.

A17: Thanks to the reviewer's suggestion! We have added the statistics of the replicates of the western blot in Figure 3e in the revised manuscript.

Q18: Page 9, Sup Fig 1a. Appreciate the western blot and agree with the interpretation, but testes is known to highly express proteins in general.

A18: Thanks to the reviewer's suggestion! In general, the genes associated with spermatogenesis are always highly or exclusively expressed in the testes. Additionally, high expression in the testicular tissue is one of the conditions for screening candidate pathogenic genes for male infertility. Therefore, here we showed that CEP128 is highly expressed in the testes, supporting the important role of CEP128 in spermatogenesis.

Q19: Page 11, lines 226-229. What are respiratory organs? What do the authors mean by no obvious ciliary defects?

A19: The respiratory organs mean the lung and trachea. No obvious ciliary defects mean that the distribution, length and the ultrastructure of the cilia in these organs from *Cep128* KO and KI mice are similar to those of WT mice. We have added this description in the revised manuscript.

Q20: Page 13, Figure 5 – No statistics, similar concern.

A20: Thanks to the reviewer's suggestion! We have added the statistics of Figure 5 in Supplementary Fig. 4 in the revised manuscript.

Q21: Page 16 – concern re: genetics for the heterozygous variants are not consistent with the findings that the homozygous variant causes infertility. 500 controls are not adequate for rare variants that may or may not be pathogenic.

A21: Thanks to the reviewer's comments! We have deleted the sentences for increased risks of infertility by heterozygous *CEP128* variants. Furthermore, another 500 normal controls for screening the heterozygous variants, and pathogenic *CEP128* variants were absent in the 1000 normal controls. We have modified this description in the revised manuscript.

Q22: Page 17, lines 350-351. What key molecules are being regulated?

A22: According to the reviewer's suggestion, we have listed the key molecules in the revised manuscript.

Q23: Page 17, line 358-359. Authors note that high levels of CEP128 hampered cilia growth which is not consistent with the title and thoughts through the paper.

A23: Here we suggested that the high expression levels of CEP128 hampered cilia growth in cultured cells, and in the title, we mean that CEP128 has no impact on cilia growth of other organs except testis *in vivo*. To avoid the confusing, we have modified the title in the revised manuscript.

Q24: Page 17, lines 360-361. The authors do not show evidence for cilia length or number in their figures and no statistics have been performed.

A24: According to the reviewer's suggestion, we have supplemented the statistics of the length and defective ultrastructure of cilia in Supplementary Fig. 5 in the revised manuscript. We are sorry that we used the inappropriate word "number" to describe the ciliary distribution, but not the real number of cilia. It is hard to count cilia number, because the sparsity of cilia is different

in various positions even in the same organ. Through HE staining, we found that the ciliary distribution in trachea, lung, eye, and brain from KO and KI mice are similar to those of WT mice. We have modified this description in the revised manuscript.

Q25: Page 18. Ciliopathies are complex disorders that affect different organ systems differently and do not necessary correlate with level of expression since almost all vertebrate cells have cilia. For example, Suppl Figure 1 shows higher levels of expression in heart and the eye, yet the patients have not phenotype.

A25: According to the reviewer's suggestion, we have deleted the improper description "we hypothesized that comparatively low CEP128 expression in the respiratory system of humans and mice might explain the normal development of respiratory cilia in the patients and mouse models". We hypothesized that cilia development in other organs might be regulated or compensated by other centrosomal proteins to allow ciliogenesis to proceed despite the lack of CEP128.

Q26: It would be helpful if in the introduction and discussion that spermatogenesis was explained so the reader who is not an expert in male infertility could better understand.

A26: According to the reviewer's suggestion, we have explained the spermatogenesis in the introduction and discussion in the revised manuscript.

Q27: Usually there are excel data tables included of all the proteomic findings (TMT).

A27: According to the reviewer's suggestion, we have added the results of TMT in the Supplementary dataset 1 in the revised manuscript.

Q28: Did the authors look at TGF beta/BMP signaling since CEP128 has been shown to be involved (PMID 29514088).

A28: According to the reviewer's suggestion, we have performed quantitative phosphoproteomics analysis on WT/KO/KI mouse testes. Notably, we found TGF- β /BMP signaling members (including Lefty2, Neo1, Fbn1, Rbl1, Gata4 and Trim33) exhibited greatly reduced phosphorylation in KI mice compared to WT mice. Moreover, Rbl1, Gata4 and Trim33 have been shown to be involved in the regulation of spermatogenesis: Rbl1 participates in the control of germ cell proliferation, differentiation and apoptosis; Gata4 cKO mice showed decreases in the quantity and motility of sperm; the testicular expression pattern of Trim33 indicate a possible involvement of Trim33 in spermatogenesis. In addition, the phosphorylation of several other non-TGF- β /BMP signaling proteins, which are associated with spermatogenic process, was also significantly decreased in KI mice, such as Fsip2, Cfap251, Akap3, Akap4, Cep131 and Odf2. No significant difference in phosphorylation of TGF- β /BMP signaling components was detected in KO mice compared to WT mice, while some proteins involved in spermatogenesis exhibited obviously reduced phosphorylation in KO mice compared to WT mice, including Fsip2, Cfap251, Prm1, Hfm1, Fam170b, Wipf3 and Top2a. Collectively, phosphorylation of TGF- β /BMP-signaling members might constitute a pivotal target for CEP128 functioning in spermatogenesis. We have supplemented these results in Figure 7f and Supplementary Fig. 9 and 10.

Minor Suggestions

Q1: Abstract – remove “could” because it is not definitive.

A1: According to the reviewer's suggestion, we have removed this word in the revised manuscript.

Q2: Introduction – page 4. Centrosome has multiple functions.

A2: According to the reviewer's suggestion, we have modified this description in the revised manuscript.

Q3: Page 5, lines 91-93 are redundant with the previous sentences.

A3: According to the reviewer's suggestion, we have simplified this description in the revised manuscript.

Q4: Page 6, line 121, remove "remarkably".

A4: According to the reviewer's suggestion, we have removed this word in the revised manuscript.

Q5: Page 7 – line 128, change to confirmed homozygosity for the CEP128 variant. The parents are heterozygous for the variant.

A5: We have added this description in the revised manuscript according to the reviewer's suggestion.

Q6: Page 7, line 141, remove "strikingly?"

A6: According to the reviewer's suggestion, we have removed this word in the revised manuscript.

Q7: Page 10, line 200, change "irrespective" to "does not play a role"

A7: According to the reviewer's suggestion, we have changed "irrespective" to "does not play a role" in the revised manuscript.

Q8: Page 11, line 233, change obtaining to "to generate"

A8: According to the reviewer's suggestion, we have changed "obtaining" to "to generate" in the revised manuscript.

Reviewer 2's comments:

Q1: In this study, a homozygous missense mutation of the CEP128 gene was identified in two siblings with primary male infertility related to cryptozoospermia from a consanguineous family through whole-exome sequencing. The mutation was then studied in two systems with three approaches: overexpression in tissue culture cells and KI in mice, and KO in mice. In tissue culture cells, the mutant CEP128 appears overexpressed, forms aggregates, abnormal cilia, nuclei, and possibly affects ceps and transcription levels. All this suggests the mutant protein has a gain of function mutation that has a toxic effect on the NC cells. Mouse model harboring the orthologous missense variant via CRISPR-Cas9 gene editing, knock-in (KI) mice, exhibited reduced sperm counts and spermatozoa with morphological abnormalities, and male infertility. Analysis of the KO CEP128 finds a milder phenotype than the KI, suggesting that the function phenotype's gain is more severe than that of the LOF phenotype. Analysis of zygote produced by ICSI fertilization with the KI and KO mutant shows that the abnormal sperm cannot support embryo development. The paper provides convincing evidence that the missense mutation of the CEP128 is a pathological mutation. The mutation is a gain of function, and it has a phenotype similar in its specific effect on sperm but different from the loss of role in several cell biological details. Potentially the particular impact is because CEP128 is highly enriched in the testis compared to other cell types.

A1: We thank the reviewer for the expert comments of our work !

Q2: The paper shows that Cep128 plays a crucial role in spermatogenesis but is not essential for viability or cilia in general. The reasoning for the tissue difference is unclear, considering that

CEP128 is thought to be a general centriole appendage protein. Also, the role of CEP128 in spermatogenesis is unknown, and it is not clear if it purely due to its centriolar function. Why the overexpression of WT-CEP128 or mutant CEP128 leads to the altered expression and toxic phenotype is unclear? Why the KI and KO mutant fertilized embryo failed to develop is unknown?

A2: We thank the reviewer for expert comments. In this study, we analyzed the cilia development of lung, trachea, brain and eye in *Cep128* KI and KO mice, and found that the distribution, length, and ultrastructure of the cilia in these organs are normal. Similarly, a previous study showed that cilia throughout *Cep131* null mice are functionally normal, even CEP131 is a general centriolar satellites protein, while the *Cep131* null mice exhibit the abnormal flagellogenesis, resulting in male infertility (Hall, E. A. et al. Acute versus chronic loss of mammalian Azi1/Cep131 results in distinct ciliary phenotypes. PLoS Genet 2013; 9, e1003928). Therefore, the reason that *CEP128* is not essential for cilia development of other organs (except the testis), might be that other centrosomal proteins but not CEP128 may play the key roles in ciliogenesis in these organs, or a compensation mechanism exists to allow ciliogenesis to proceed despite the lack of CEP128.

In this study, deleterious *CEP128* variants impaired spermatogenesis in humans and mice, we thus suggest that *CEP128* plays an essential role in spermatogenesis. We found the abnormal centrioles in testes in humans and mice carrying homozygous *CEP128* variants, so we speculated that the aberrant CEP128 expression results in defects in centrioles, considering CEP128 is a centriole appendage protein. Because centrioles play important roles in spermatogenesis, the defective centrioles resulting from homozygous *CEP128* variants would contribute to the impaired spermatogenesis. In addition, we detected the reduced expression in genes/p-proteins associated with male reproduction in *Cep128* KI and KO mice by proteomics approach, RNA-seq and quantitative phosphoproteomics analysis. Therefore, we suggested that the defective centrioles and the abnormal expression of molecules related to male fertility cooperatively disrupt the process of spermatogenesis.

In this study, we found that overexpression of WT-CEP128 or mutant CEP128 leads to short or no cilia. Importantly, CEP128 has been suggested to mediate primary ciliogenesis via regulating centriolin, NIN, and the NIN group members expressing on the centriole subdistal appendages in hRPE-1 cells (1. Mazo et al. Spatial Control of Primary Ciliogenesis by Subdistal Appendages Alters Sensation-Associated Properties of Cilia. *Dev Cell* 2016; 39, 424-437. 2. Kashihara et al. Cep128 associates with Odf2 to form the subdistal appendage of the centriole. *Genes Cells* 2019; 24, 231-243). Therefore, we examined the expression of these well-established centrosomal proteins in cells transfected with the WT-CEP128 and CEP128^{R222Q} plasmids as well as NC cells, and the decreased protein levels of NIN, centriolin and CEP170 were detected in cells overexpressing WT-CEP128 and CEP128^{R222Q}. We thus speculated that the WT-CEP128 or mutant CEP128 may inhibit cilia development by downregulating the expression of NIN, centriolin and CEP170.

Through proteomics approach and RNA-seq, we detected several genes related to fertilization are reduced in KO and KI male mice, we thus suggested that the impaired fertilization observed in KO and KI mice might be associated with the reduced expression of Crisp1, Wbp2nl, Tmem95, Eqtn and Calr3, which are involved in gamete fusion and egg activation mediated by meiotic resumption and pronuclear development (1. Da Ros, V. G. et al. Impaired sperm fertilizing ability in mice lacking Cysteine-Rich Secretory Protein 1 (CRISP1). *Dev Biol* 2008; **320**, 12-18. 2. Castillo, J., Jodar, M. & Oliva, R. The contribution of human sperm proteins to the development and epigenome of the preimplantation embryo. *Hum Reprod Update* 2018; **24**, 535-555,. 3. Azad, N. et al. Oligoasthenoteratozoospermic (OAT) men display altered phospholipase C zeta (PLCzeta) localization and a lower percentage of sperm cells expressing PLCzeta and post-acrosomal sheath WW domain-binding protein (PAWP). *Bosn J Basic Med Sci* 2018; **18**, 178-184. 4. Wu, A. T. et al. PAWP, a sperm-specific WW domain-binding protein, promotes meiotic resumption and pronuclear development during fertilization. *J Biol Chem* 2007; **282**, 12164-12175. 5. Lamas-Toranzo, I. et al. TMEM95 is a sperm membrane protein essential for mammalian fertilization. *Elife* 2020; 9. 6. Hao, J. et al. Equatorin is not essential for acrosome biogenesis but is required for the acrosome reaction. *Biochem Biophys*

Res Commun 2014; 444, 537-542. 7. Ikawa, M. *et al.* Calsperin is a testis-specific chaperone required for sperm fertility. *J Biol Chem* 2011; **286**, 5639-5646.)

Q3: The paper has many interesting findings that can be further developed but appear to have a critical mass of information that is likely to make it attractive to Nature Communication's audience once all the minor points below are addressed.

A3: We thank the reviewer to approve our work, and we have addressed all the minor points below in the revised manuscript.

Minor points:

- Unconventional description/statements:

Q1: "centrosomal protein (CEP) family"- centrosomal protein (CEP) are not family of protein – they are functional or structure associated group.

A1: We are sorry for bringing the doubts. In the previous manuscript, we want to state "CEP family protein", which are characterized as CEP × kDa proteins, but not the general centrosomal proteins. Considering the whole story of this manuscript, we think it is better to state the general centrosomal proteins, we thus have modified the "centrosomal protein (CEP) family" to "centrosomal proteins" in the revised manuscript.

Q2: "33 CEPs have been identified" – this need reference and detention as many will claim that there are hundreds of centrosomal proteins.

A2: Similarly, "33 CEPs" here means the "CEP × kDa proteins", but not the "general centrosomal proteins". We have changed the "CEP × kDa proteins" into "general centrosomal proteins" and also rewritten this part in the revised manuscript.

Q3: “few are related to microcephaly⁴.”- I would say surprisingly, several of them are related to microcephaly.

A3: We are sorry for this typo. It should read “a few are related to microcephaly”. We have corrected it in the revised manuscript.

Q4: “dysplasia of the fibrous sheath (DFS) and globozoospermia is always accompanied by centrosome dysfunction⁷” - I would say in many – not always

A4: According to the reviewer’s suggestion, we have changed “always” to “in many” in the revised manuscript.

Q5: There is a need to provide information on the role of Cep128 in the centriole as a subdistal appendage.

A5: In the previous manuscript, we described the role of Cep128 in the centriole as a subdistal appendage in the Discussion. According to the reviewer’s suggestion, we have supplemented the description about “the role of Cep128 in the centriole as a subdistal appendage” in the introduction of the revised manuscript.

Q6: “Initially, no spermatozoa were observed in the replicate wet preparations” – what is wet preparations? Is it a semen sample?

A6: Yes, the wet preparation is a semen sample, which is used to assess the sperm count preliminarily.

Making a preparation:

- Mix the semen sample well.
- Remove an aliquot of semen immediately after mixing, allowing no time for the spermatozoa to settle out of suspension.
- Remix the semen sample before removing replicate aliquots.
- Place a standard volume of semen (10 μ l) onto a clean glass slide.
- Cover it with a coverslip (22 mm \times 22 mm) to provide a chamber approximately 20 μ m deep. The weight of the coverslip spreads the sample.
- Take care to avoid the formation and trapping of air bubbles between the coverslip and the slide.
- Assess the freshly made wet preparation as soon as the contents are no longer drifting.

Q7: The paper's main text should start with Fig 1 and not Fig 2.

A7: According to the reviewer's suggestion, we have modified the order of Fig 1 and Fig 2 in the revised manuscript.

Q8: Fig. 1 – please define in the legend all the elements in the figure, including III-3 and III-2

A8: We thank the reviewer for kind suggestions! We have defined all the elements in the legend of Fig 1 in the revised manuscript.

Q9: Fig. 2c – it is essential to have similar picture orientations in the normal control and mutants – without that, it is hard to assess the sperm connecting piece phenotype. Also - reorient the individual panels, so they all have the same orientation for each subcategory. Mark

the location of PC and DC in each panel and describe their phenotype. Also, label the mitochondria.

A9: According to the reviewer's suggestion, we have reoriented the panels and marked the location of PC and DC and described their phenotype, and also labeled the mitochondria (M) in Fig 2 in the revised manuscript.

Q10: Fig 2b "The site of the CEP128 missense mutation is evolutionally conserved among various species" – please explain if they are all mammals – what happened to the amino acid in more distinct evolutionary groups like insects and worms?

A10: Thanks to the reviewer's comment! Through complete alignment by ClustalX, we found that this site is a conservative Arginine site in multiple species, including mammals, amphibians, fish, birds, and invertebrates, such as Mouse, *Xenopus laevis*, Chicken, Nile tilapia and *Millepora damicornis*. Therefore, this site is conserved from lower to higher organisms and is not only limited to mammals. However, we did not find the CEP128 protein sequence of insects or worms in the Uniprot database. We have made changes to Figure 2b.

Q11: "the spermatozoa from the two patients represented the lack of the central-pair microtubules (CPs)". One of the patients has clear central-pair microtubules – please correct the sentence.

A11: We thank the reviewer for kind reminders! We have corrected this sentence in the revised manuscript.

Q12: Fig. 2d – is the CEP128 localize to the distal or proximal centriole?

A12: We are sorry that we showed the poor quality of Fig. 2d in the previous manuscript, and it is hard to identify the exact localization of CEP128 in sperm centriole. To precisely identify the

location of CEP128 in sperm centrioles, we used anti-acetylated-tubulin antibody to label DC and PC, and modified the method of Immunofluorescence staining according to Fishman *et al.* (A novel atypical sperm centriole is functional during human fertilization. *Nat Commun.* 2018; 9(1):2210), which has been shown in the method part of the revised manuscript. The results of immunofluorescence staining showed that CEP128 colocalizes with both PC and DC in the sperm of normal control. We have supplemented the results in Fig. 2b of the revised manuscript.

Q13: “With the accumulation and ectopic expression of EGFP-CEP128 mutant in the cytoplasm short cilia” show examples of short cilia – this phenotype needs quantification and statistical analysis.

A13: According to the reviewer’s suggestion, we have supplemented the quantification and statistical analysis of the phenotype of short cilia in Fig 3b of the revised manuscript.

Q14: 1a-c and Fig 2 a, b, c, and many other figures and descriptions (such as nuclear morphology) need quantification and statistical analysis.

A14: According to the reviewer’s suggestion, we have supplemented the quantification and statistical analysis to the figures and description where need in the revised manuscript.

Q15: Fig 3b needs insets with zoon on basal body and cilia.

A15: According to the reviewer’s suggestion, we have inserted the picture of basal body and cilia in Fig 3b of the revised manuscript.

Q16: Define Pc

A16: Pc is defined as proximal centriole. In the previous manuscript, we defined Pc in on Page 6 line 110. In the revised manuscript, we have defined the PC in the legend of Figure 2.

Q17: What is “sperm ultrastructural disarray”?

A17: “sperm ultrastructural disarray” means the abnormal arrangements of axoneme. We have added this description in the revised manuscript.

Q18: The paper legends are poorly written and need extensive work.

A18: We thank the reviewer for helpful comments. We have carefully revised the manuscript before re-submission. The English of the manuscript text has also been polished by a professional language editing.

Q19: “ICSI failed in these patients because of the aberrant proximal centrioles exhibited in almost all the spermatozoa” – add an explanation on the possible effect on distal atypical centriole.

A19: Thanks to the reviewer’s kind suggestion! Actually, the distal atypical centriole is released into the zygote, nucleates a daughter centriole and participates in spindle pole formation during fertilization (Avidor-Reiss T, Fishman EL. It takes two (centrioles) to tango. *Reproduction*. 2019 Feb;157(2):R33-R51.). Thus, the spermatozoa’s distal atypical centriole plays an important role in the zygote development. In the previous manuscript, we only labeled the defective proximal centrioles in the spermatozoa of the normal control and patients, so here we only presented the aberrant proximal centrioles might result in the failed ICSI. According to the reviewer’s suggestion, we have labeled the distal atypical centrioles on the sperm of the control and patients, and rewritten this sentence into “ICSI failed in these patients because of the aberrant centrioles exhibited in almost all the spermatozoa” in the revised manuscript.

Reviewer 3's comments:

Q1: The manuscript by Ying Shen and colleagues presents convincing evidence for a role of CEP128 in human and mouse spermatogenesis, and for a mutant (R222Q) probably causing male infertility in human. Overall, the manuscript is well written and the figures are very nice. The results are novel and potentially important in terms of male infertility diagnosis and treatment. The mechanistic part of the study is less convincing. It lacks important methodological details and intermediate results, and the logic is sometimes difficult to follow.

A1: We thank the reviewer to approve our work! According to the reviewer's suggestion, we have added a quantitative phosphoproteomics analysis on *Cep128* KO/KI/WT mouse testes to decipher the potential molecular mechanisms of CEP128 function in spermatogenesis.

Please find below a few suggestions for improvement:

Q1: In contrast to what is mentioned on Page 6 lines 123-126 ("This mutation is absent in all of the human population genome databases"), this mutation is already described in GnomAD (<https://gnomad.broadinstitute.org/variant/14-81329198-C-T>) and dbSNP (rs114668642). Please revise the text accordingly.

A1: We thank the reviewer for kind reminder. The updated data showed that this variant is absent in African, Europe, South Asian and American populations and at a low frequency in Asians in ExAC Browser, 1000Genomes and gnomAD databases. We have corrected this error and added these data in supplementary table 1 in the revised manuscript.

Q2: The title "Dysfunction of CEP128 impairs spermatogenesis but not ciliogenesis" is a bit confusing: Fig 3 confirms that CEP128 is involved in the formation of cilia in RPE cell lines. In addition, the flagellar defects observed in CEP128 KO/KI mice also suggest that CEP128 is

involved in flagella biogenesis. The presented results suggest that CEP128 dysfunction does not impair ciliogenesis in respiratory organs, but do not exclude effects on ciliogenesis in other structures.

A2: Thanks to the reviewer's comment! Our observation that over-expression of the mutant protein in RPE cells resulted in short or no cilia confirmed the deleteriousness of this mutation *in vitro*. *In vivo*, we found that deleterious *CEP128* variants could impair spermatogenesis in humans and mice, but the cilia of lung, trachea, brain and eye from mouse models were normal and the patients carrying homozygous *CEP128* p.R222Q variant presented that they had no other phenotypes except infertility. Furthermore, a previous study showed that another centrosomal protein CEP131 (which is a different protein from CEP128 in this study) localizes to centriolar satellites, and loss of CEP131 impairs ciliogenesis in mouse fibroblasts *in vitro*. However, cilia throughout *Cep131* null mice are functionally normal, as embryonic patterning and adult homeostasis are grossly unaffected. Surprisingly, the *Cep131* null mice exhibit the abnormal flagellogenesis, resulting in male infertility (Hall, E. A. et al. Acute versus chronic loss of mammalian Azi1/Cep131 results in distinct ciliary phenotypes. *PLoS Genet* 2013; 9, e1003928). Therefore, we suggested that *CEP128* p.R222Q variant perturbing cilia growth in cultured cells but not the cilia development of other organs (except testis) *in vivo*, because the regulation mechanism of cilia growth *in vivo* is more complex and other centrosomal proteins but not CEP128 may play the key roles in ciliogenesis in these organs, or a compensation mechanism exists to allow ciliogenesis to proceed despite the lack of CEP128. The manuscript has been revised accordingly.

In the previous title, we said that "Dysfunction of CEP128 impairs spermatogenesis but not ciliogenesis in humans and mice". This title is a bit confusing, because the flagellum is a special cilium. Therefore, we have also modified the title in the revised manuscript. We have added the results of normal ciliogenesis in other structures including eye and brain in Supplementary Fig. 5 in the revised manuscript.

Q3: The expression of CEP128 is clearly downregulated (but not absent; this should be stressed in the text) in testes of KO mice and upregulated in testes of KI mice (qPCR and WB). CEP128 expression should also be checked in oocytes and lung of KO and KI mice before concluding that CEP128 does not play any role in these tissues (page 10 line 200, page 11 line 227).

A3: In this study, we microinjected CRISPR-Cas9 reagents into mouse zygotes to generate the *Cep128* KO and KI mice. To mimic mutation in men, we chosen the corresponding mouse exon 10 of *Cep128* for gene editing. KO mice with a frameshift mutation (5 bp deletion) and KI mice with human point mutation were selected for further analysis. The 5 bp deletion mutation is predicted to cause premature translational termination. In addition, the remaining mutant *Cep128* mRNA and protein were significantly lower than those in WT mice, suggesting the decay of mutant *Cep128* mRNA and protein. We did not use conditional gene knockout/knockin system, so the altered CEP128 expression is similar in all the tissues of KO or KI mice. We suggested that CEP128 might not play a role in ciliogenesis in other organs except testis, and in these tissues, other centrosomal proteins but not CEP128 may be responsible for cilia growth. According to the reviewer's suggestion, we have checked CEP128 expression of oocytes and lung from KO and KI mice, and the results have been showed here (Fig R2a).

Figure R2. The expressions of CEP128 in mouse lung and oocyte (a), and in human testis (b).

Q4: A large part of the presented results relies on immunofluorescence analysis of human and mice samples performed with HPA001116 antibody. This antibody has already been used in human cell lines, but I could not find any validation study in human sperm, or in any mouse sample. The authors should provide at least a full-range western blot showing a single band at the expected MW in human sperm and in sperm from WT, KO and HI mice. They should also show that HPA001116 reproduces the results obtained with the Flag antibody on Figure 3a.

Page 7 line 132: please precise what “markedly enhanced” means. Could you provide any quantitative measurement of the signal?

A4: According to the reviewer’s suggestion, we have provided a full-range western blot showing a single band at the expected MW in human sperm in the above Figure R2b. Actually, the HPA001116 antibody has already been used in mouse testes of WT, KO and KI in Fig. S2d of our original submission (in Fig. S3d in the revised manuscript), and the full-range western blot has been shown in supplementary information in the revised manuscript. For mouse mature sperm, there is no centriole, so we did not investigate CEP128 expression of mouse sperm. We also have used HPA001116 antibody to reproduce the results obtained with the Flag antibody on Figure 3a, which has been showed in Figure 3c in the previous manuscript. Additionally, we have provided the quantitative measurement of the enhanced signal of mutant CEP128 protein by image J in Figure 3a of the revised manuscript.

Q5: Please provide details about the CEP128 construct used for cell line transfection (either the full sequence or the database accession number of “wild type CEP128”).

A5: According to the reviewer’s suggestion, we have provided the database accession number of “wild type CEP128” in the method part in the revised manuscript.

Q6: The list of the 399 proteins shown to be differentially expressed by the proteomics approach should be provided, with fold changes. How many proteins were up/down in KO and KI mice? What fold change and p value were considered for a differential expression? The choice of the five proteins to follow up appears quite subjective. Were those proteins the ones with the highest fold change? Why not choosing all the ones presented in the Reproduction diagrams Fig 7b/c ? Or the ones also found with the RNA analysis?

A6: 59 down-regulated proteins and 113 up-regulated proteins were detected in *Cep128* KO mice compared to WT mice. 100 down-regulated proteins and 127 up-regulated proteins were

detected in KI mice compared to WT mice. “1.2-fold change” and “ $p < 0.05$ ” were considered for a differential expression in this study. Different from label free, TMT has compression effect. The reason is that TMT uses the intensity of secondary reporting ions for quantification. However, when parent ions are isolated, some co-elution and co-fragmentation peptides are mixed in the parent ions, which will interfere the results of reporting ions, thus resulting in the compression effect. Therefore, the 1.2-fold change is required.

Protein quantification calculation method: In this project, the quantitative values of each sample were obtained through repeated experiments of total protein quantification. The first step is to calculate the differential expression of proteins between the two samples: Initially, calculate the average quantitative value of each sample in multiple repetitions, and then calculate the ratio of average values between two samples, and this ratio is as the final differential expression of the comparison group. The second step is to calculate the significant p-value of different expression of each protein in the two samples: calculate the log₂ value of relative quantitative value of each sample to make the data conform to normal distribution, and then the p-value is calculated by two-sample two-tailed test. According to the reviewer’s suggestion, we have added the list of proteomics results in Supplementary dataset 1 in the revised manuscript.

The differential proteins presented in the Reproduction diagrams Fig 7b/c are the ones highly or strictly expressed in testis, or the Gene Ontology or KEGG pathway analysis indicate that ones might be involved in spermatogenesis. Among these differential proteins, we choose Wbp2nl, Rcbtb2, Prss55, Crisp1 and Defb22 as the candidate proteins, because the five proteins have been substantiated to play the important roles in male reproduction in the mouse/rat models or infertile patients: the sperm cells expressing Wbp2nl was significantly lower in oligoasthenoteratozoospermic patients compared to control group; RC/BTB2 is suggested to play a role in transporting proteins during acrosome formation in spermatogenesis; male mice lacking *Prss55* gene show severe fertility defects; *Crisp1*^{-/-} male mice exhibit a significantly reduced sperm fertilizing ability; Defb22 plays a crucial role in sperm production. Therefore, we confirmed the decreased expression of these proteins in *Cep128* KI or KO mice by western blot

and immunofluorescence staining, although those proteins were not with the highest fold change.

For the results of RNA-seq, we also choose several differential genes for further confirmation by qPCR in Supplementary Fig. 8a-c, which have been suggested to be essential for spermatogenesis: *Spata31*-deficient male mice exhibited low sperm count and premature shedding of germ cells into the lumen, ultimately causing azoospermia and male sterility; *Ccdc63* removal resulted in sterile male mice due to shortened flagella; *Tssk4* KO male mice were subfertile due to seriously decreased sperm motility associated with the defective ultrastructure of sperm flagellum; male *Cdo1*^{-/-} mice exhibit idiopathic infertility owing to the increase in head abnormalities and the defects in post-testicular sperm maturation; *Inpp4b*^{-/-} males produced fewer mature sperm cells compared to WT; *Spata48*^{-/-} knockout male mice had smaller testis and defective spermatogenesis compared to WT mice; *Obif*^{-/-} mice show a significant decrease in testis weight as well as in sperm number; analysis of abnormal expression in infertile male patients revealed complete absence of NYD-SP16 in the testes of patients with Sertoli-cell-only syndrome and variable expression in patients with spermatogenic arrest; *TMEM95*-deficient sperm were unable to fuse with the egg membrane or penetrate into the ooplasm; *Insl5*^{-/-} mice displayed impaired male fertility that is due to marked reduction in sperm motility; *KLC3* transgenic males have a significantly reduced sperm count and produce spermatozoa that exhibit abnormal motility parameters; *Cst8*^{-/-} male mice showed abnormally shaped sperm heads and tails were noted along with immature germ cells; subfertility and oligozoospermia were noticed in such animals with low *Wnt3* expression in post-pubertal Sertoli cells; *LRRC52* KO results in mice with severely impaired fertility; *Eqtn*^{-/-} mice presented dramatically reduced fertilization and acrosome exocytosis rates; *Calr3*^{-/-} males produced apparently normal sperm but were infertile because of defective sperm migration from the uterus into the oviduct and defective binding to the zona pellucida; *Sox30*-null mice represent a complete arrest of spermatogenesis at the onset of spermiogenesis; premature translation of *Tnp2* mRNA in male mice caused abnormal head morphogenesis, reduced sperm motility and male infertility; *SUN5* is the causative gene of acephalic spermatozoa syndrome in both humans and mice; *IRS-4* null mice reproduced less litters than wild-type mice; the abnormal sperm

morphology and sperm chromatin condensation are related to decreased *CatSper* gene expression in mice.

Q7: The methodological details about the RNA sequencing of mouse testes and the differential expression analysis are missing. Please provide the full lists of the differentially regulated genes, with fold changes. How do these lists compare with the lists obtained with the proteomics approach? How do the five candidates selected above behave at transcript level?

A7: According to the reviewer's suggestion, we have added the methodological details about the RNA sequencing of mouse testes in the Methods part and provided the full lists of the differentially regulated genes in Supplementary dataset 2 in the revised manuscript. Among the five candidates selected by proteomics approach, we only detected the reduced mRNA levels of *Defb22* in *Cep128* KO and KI mice, and for *Wbp2nl*, *Rcbtb2*, *Prss55* and *Crisp1*, the increased ubiquitination-mediated degradation of these proteins was observed in *Cep128* KO and KI mice compared to WT mice. Therefore, the decreased protein level of *Defb22* is associated with transcriptional regulation, and the diminished protein levels of *Wbp2nl*, *Rcbtb2*, *Prss55* and *Crisp1* are related to post-translational modification. It is usually believed that there is some correlation between mRNA and protein levels. However, many proteomics and genomic studies have reported that the correlation between mRNA level and protein amount is poor, and the average distribution of correlation coefficient is about 0. This may involve many biological mechanisms, and these mechanisms are not mutually exclusive: (1) It may be attributed to the post-translational modification of proteins, and the secondary structure of mRNAs may also be modified (such as m6A methylation modification); (2) The degradation rate of mRNAs is different from that of proteins, and half-life between proteins is also various; (3) Many transcripts can be produced by alternative splicing of exons in transcriptomes, but most may not be translated into proteins. Moreover, we use Data Dependent Acquisition (DDA) to obtain the proteomics data. In this mode, the top 20 peptides are identified, that means the 20 peptides with the highest signal strength are further for secondary fragmentation. Therefore, if the peptide abundance does not reach the top 20, the mass spectrum will not be able to

conduct secondary fragmentation on these peptides, and these proteins cannot be identified. Therefore, we combined the results of proteomics and RNA-seq to investigate the key molecules which might contribute to male infertility in *Cep128* KO and KI mice. Of course, uncovering the exact mechanism of CEP128 regulating reproductive process needs more future research.

Q8: The Co-IP results are interesting but rather preliminary. A putative link between the formation of the complex and ubiquitination-mediated degradation of the members of the complex is missing. Is one of the complex members known to be involved in ubiquitination-dependent pathways? Did you identify proteins involved in ubiquitination pathways in your differential expression analyses?

A8: Thanks to the reviewer's expert comment! In this study, we found that several proteins were decreased in male *Cep128* KI or KO mice compared to WT mice. Consequently, we wondered whether CEP128 decreased these proteins by transcriptional levels or translational levels. The limited difference in the mRNA levels of *Wbp2nl*, *Rcvtb2*, *Prss55* and *Crisp1* between the *Cep128* KO/KI and WT mice indicates that CEP128 may not affect their transcription. Considering that ubiquitination-mediated degradation is a common mechanism for protein degradation, so we investigated the differential ubiquitination of these proteins in KI, KO and WT mice to simply explore the protein degradation mechanism of these differential expression proteins in KI and KO mice. However, no study has suggested any information about ubiquitination of *Wbp2nl*, *Rcvtb2*, *Prss55* and *Crisp1*. In addition, E3s are the most heterogeneous class of enzymes in the ubiquitination pathway, and there are so many E3s in mice. We also did not identify any significant proteins involved in ubiquitination pathways in our differential expression analyses. Therefore, we only investigate the differential ubiquitination of these proteins by Co-IP. But the reviewer's suggestion is meaningful, and we will completely elucidate the mechanism of CEP128 regulating the gene expression in future study, considering the important role of CEP128 in spermatogenesis.

Q9: A recent study by Mönnich et al. (29514088) suggested that CEP128 affects TGF- β 1-induced phosphorylation of multiple proteins that regulate cilium-associated vesicle trafficking. Would this be compatible with your results? A quantitative phosphoproteomics analysis of WT/KO/KI mouse testes might help deciphering the molecular mechanisms of CEP128 function in spermatogenesis.

A9: According to the reviewer's suggestion, we have performed quantitative phosphoproteomics analysis on *Cep128* KO/KI/WT mouse testes. Notably, we found TGF- β /BMP signaling members (including Lefty2, Neo1, Fbn1, Rbl1, Gata4 and Trim33) exhibited greatly reduced phosphorylation in KI mice compared to WT mice. Rbl1, Gata4 and Trim33 have been shown to be involved in the regulation of spermatogenesis: Rbl1 participates in the control of germ cell proliferation, differentiation and apoptosis; Gata4 cKO mice showed decreases in the quantity and motility of sperm; the testicular expression pattern of Trim33 indicate a possible involvement of Trim33 in spermatogenesis. In addition, the phosphorylation of several other non-TGF- β /BMP signaling proteins, which are associated with spermatogenic process, was also significantly decreased in KI mice, such as Fsip2, Cfap251, Akap3, Akap4, Cep131 and Odf2. No significant difference in phosphorylation of TGF- β /BMP signaling components was detected in KO mice compared to WT mice, while some proteins involved in spermatogenesis exhibited obviously reduced phosphorylation in KO mice compared to WT mice, including Fsip2, Cfap251, Prm1, Hfm1, Fam170b, Wipf3 and Top2a. Collectively, phosphorylation of TGF- β /BMP signaling might be a pivotal target for CEP128 to function in spermatogenesis. We have supplemented these results in Figure 7f and Supplementary Fig. 9 and 10.

If there are any more questions, please do not hesitate to let us know. We are willing to explain it. Thank you very much!

Sincerely,

Ying Shen, Ph. D.

Professor, Department of Obstetrics/Gynecology, Key Laboratory of Obstetric, Gynecologic and Pediatric Diseases and Birth Defects of Ministry of Education

West China Second University Hospital, Sichuan University

P. R. China

REVIEWER COMMENTS

Reviewer #2 (Remarks to the Author):

The paper is vastly improving and is well suitable for Nature Communication.

Few minor comments:

- The acronym WES is used only four times in the text, and I suggest spelling it out for clarity.
- “while CEP128 signals were markedly enhanced in the sperm neck of the patients with the loss of centriole staining (Fig. 2d).” – confusing – please clarify “centriole staining” – is that tubulin staining? This “centriole staining needs quantification.
- Please explain “pyknosis” in the text and mark it in the figure.
- Negative control (NC) need to be defined in the figure legend.
- In the absence of a centriolar marker, the location is inconclusive. Please change to “Immunofluorescence staining revealed that CEP128 was detectable in the centrioles or their vicinity of various germ cells, except the steps 15-16 and the mature sperm (Supplementary Fig. 207 2c).” i.e., add “or their vicinity”.

Tomer Avidor-Reiss

Reviewer #3 (Remarks to the Author):

The manuscript has improved a lot and the inclusion of the phosphoproteomics data further increases its value.

I still have some concerns with the experiments presented in Supplementary Fig. 7c-g, which lack essential controls. If it is not possible to provide those controls, I would recommend to remove them from the manuscript.

More specifically:

- 1) Supplementary Fig. 7c

In order to justify that “CEP128 could bind to Wbp2nl, Rcbtb2, Prss55 and Crisp1”, the following items should be shown:

- Enrichment of CEP128 in the IP compared to the input
- A protein not copurifying with CEP128 in the IP
- Results of a Co-IP performed using an unrelated antibody or a preimmune serum

2) Supplementary Fig. 7d-g:

Methodological details are lacking. I presume that Wbp2nl, Rcbtb2, Prss55 and Crisp1Abs were used for the blots entitled “IP:Ubiquitin”, but this should be specified.

Unfortunately the “full scans” appended at the end of the Supplementary file are not full range. Do you observe any high MW bands using Wbp2nl, Rcbtb2, Prss55 and Crisp1Abs when analyzing KO/KI extracts by WB (prior IP ubiquitin)?

This experiment would be more convincing if you could show that this increase of ubiquitination has some specificity towards your proteins of interest. At least, an additional panel showing that Gapdh ubiquitination is not affected in KO/KI mice would be required.

Minor:

Abstract: “centrosomal proteins are necessary for the components of the centrosome.” Do you mean “centrosomal proteins are necessary components of the centrosome.”?

Reviewer #4 (Remarks to the Author):

This revised version of the work by Zhang and colleagues provides convincing data indicating that disrupted CEP128 function affects spermatogenesis. However, there are specific issues that still require attention and should be taken into account by the authors.

(1) The authors do not properly report the population frequency data for the identified CEP128 missense variant (rs114668642). Population data indicate that MAF is 0.0234 and 0.0113 in the Japanese and Korean populations (see https://www.ncbi.nlm.nih.gov/snp/rs114668642?horizontal_tab=true#frequency_tab; 3KJPN and KRGDB studies). The authors statement (page 7, lines 143-144) and Suppl. Table 1 apparently do not take into account such information and do not reflect the real picture. Text and Table should be revised taking in to account all the available population data;

(2) The authors do not provide any information on the data output of the genomic analysis. This is a key aspect, particularly considering the consanguinity of the family. How many rare/private clinically relevant variants were annotated? How many rare functionally relevant homozygous variants? Did the authors exclude the possibility of other contributing cis/trans-acting events? Did the authors exclude absence of coverage (i.e. possible homozygous deletion) in coding exons of genes implicated in male infertility? Did the authors perform a SNP array analysis to exclude relevant LoH regions?

(3) Table 3 and main text (page 17 line 369 to page 18 line 376): it is not clear the genotype of the reported cases. Are they heterozygous or homozygous for the CEP128 changes? This is a relevant aspect;

(4) The authors do not report the phenotype of the heterozygous KI mice. Presence/absence of relevant signs/features related to spermatogenesis/fertility should be provided. This is an important issue since the authors provide in vitro data apparently suggesting a “dominant” role of the R222Q CEP128 mutant. On the other hand, the homozygous condition in patients would suggest LoF, which is also in line with the "documented" fertility of the father.

Other remarks:

(5) Title: as previously noted by reviewer 1, the title does not reflect the major findings of this work. Based on the in vivo data, the authors may consider to revise the title to emphasize that “Disrupted/altered (or Loss of) CEP128 function impairs...”;

(6) Abstract (line 49): the identified variant is an annotated variant (rs114668642) (i.e., is not “novel”). The authors should revised the text accordingly.

(7) Abstract (line 52) and throughout the main text: the authors should specify that the KI mice are homozygous for the missense change;

(8) From the pedigree reported in figure 2A, it is not clear the level of consanguinity. The authors should provide a complete pedigree of the family.

(9) Results/Tables: the authors should add the SNP IDs for the identified CEP128 variants.

Reviewer #2 (Remarks to the Author)

The paper is vastly improving and is well suitable for Nature Communication.

A: We thank the reviewer for helping approve our work!

Few minor comments:

Q1- The acronym WES is used only four times in the text, and I suggest spelling it out for clarity.

A1: Thanks for the reviewer's kind suggestion. We have changed "WES" into "whole exome sequencing" in the revised manuscript.

Q2- "while CEP128 signals were markedly enhanced in the sperm neck of the patients with the loss of centriole staining (Fig. 2d)." – confusing – please clarify "centriole staining" – is that tubulin staining? This "centriole staining needs quantification.

A2: We used anti-AcTubulin (ab24610, abcam) to label the centrioles according to the study of Fishman *et al.* (*Nat. Commun.* 2018;9:2210). Therefore, the tubulin staining represented the centriole staining. To avoid the confusion, we have changed "centriole staining" into "tubulin staining labeling the centriole" in the revised manuscript. In addition, we have supplemented the quantification of centriole staining in Figure 2d in the revised manuscript.

Q3- Please explain "pyknosis" in the text and mark it in the figure.

A3: Here we used "pyknosis" to describe the abnormal morphology of nucleus (smaller nucleus). We have modified this inappropriate description in the revised manuscript.

Q4- Negative control (NC) need to be defined in the figure legend.

A4: We thank the reviewer for kind suggestion. We have added the definition of NC in the figure legends in the revised manuscript accordingly.

Q5- In the absence of a centriolar marker, the location is inconclusive. Please change to “Immunofluorescence staining revealed that CEP128 was detectable in the centrioles or their vicinity of various germ cells, except the steps 15-16 and the mature sperm (Supplementary Fig. 207 2c).” i.e., add “or their vicinity”.

A5: We have changed the inappropriate description of “Immunofluorescence staining revealed that CEP128 was detectable in the centrioles of various germ cells, except the steps 15-16 and the mature sperm” into “Immunofluorescence staining revealed that CEP128 was detectable in the centrioles or their vicinity of various germ cells, except the steps 15-16 and the mature sperm” according to the reviewer’s suggestion.

Reviewer #3 (Remarks to the Author)

The manuscript has improved a lot and the inclusion of the phosphoproteomics data further increases its value.

I still have some concerns with the experiments presented in Supplementary Fig. 7c-g, which lack essential controls. If it is not possible to provide those controls, I would recommend to remove them from the manuscript.

A: We thank the reviewer for kind helps to approve our work. We have added the essential controls in Supplementary Figure 7c and 7h in the revised manuscript. Alternatively, we could remove this part if the reviewer thinks that our supplements are not appropriate.

More specifically:

1) Supplementary Fig. 7c

Q1: In order to justify that “CEP128 could bind to Wbp2nl, Rcbtb2, Prss55 and Crisp1”, the following items should be shown:

- Enrichment of CEP128 in the IP compared to the input*
- A protein not copurifying with CEP128 in the IP*

- Results of a Co-IP performed using an unrelated antibody or a preimmune serum

A1: We thank the reviewer for expert comments. We have added the “Enrichment of CEP128 in the IP compared to the input”, “GAPDH that not copurifying with CEP128 in the IP”, and “Results of a Co-IP performed using an IgG antibody as the negative control” in Figure S7c of the revised manuscript.

2) Supplementary Fig. 7d-g:

Q1: Methodological details are lacking. I presume that Wbp2nl, Rcbtb2, Prss55 and Crisp1Abs were used for the blots entitled “IP:Ubiquitin”, but this should be specified.

A1: According to the reviewer’s comment, we have added the methodological details in the legend of Figure S7d-g in the revised manuscript. Also, we have labeled “WB: Wbp2nl, Rcbtb2, Prss55 and Crisp1” in Figure S7 d-g.

Q2: Unfortunately, the “full scans” appended at the end of the Supplementary file are not full range. Do you observe any high MW bands using Wbp2nl, Rcbtb2, Prss55 and Crisp1Abs when analyzing KO/KI extracts by WB (prior IP ubiquitin)?

A2: Prior to IP ubiquitin, we observed bands using Wbp2nl, Rcbtb2, Prss55 and Crisp1Abs when analyzing mouse testis extracts by WB. We only observed another high MW band using anti-Rcbtb2 antibody. Here, we also provided the full scans of the WB results (Figure R1).

Figure R1. WB analysis of bands using Wbp2nl, Rcbtb2, Prss55 and Crisp1Abs of mouse testis extract.

Q3: This experiment would be more convincing if you could show that this increase of ubiquitination has some specificity towards your proteins of interest. At least, an additional panel showing that Gapdh ubiquitination is not affected in KO/KI mice would be required.

A3: According to the reviewer's suggestion, we performed the Co-IP of Gapdh ubiquitination in the KO/KI and WT mice, and found that the Gapdh ubiquitination showed no significant differences between KO/KI mice and WT mice. The results have been shown in Figure S7h.

Minor:

Q1: Abstract: "centrosomal proteins are necessary for the components of the centrosome." Do you mean "centrosomal proteins are necessary components of the centrosome."?

A1: We thank the reviewer for kind reminder. We have changed "centrosomal proteins are necessary for the components of the centrosome." into "centrosomal proteins are necessary components of the centrosome" in the revised manuscript.

Reviewer #4 (Remarks to the Author):

This revised version of the work by Zhang and colleagues provides convincing data indicating that disrupted CEP128 function affects spermatogenesis. However, there are specific issues that still require attention and should be taken into account by the authors.

A: We thank the reviewer's comments. We have revised our manuscript accordingly.

Q1: The authors do not properly report the population frequency data for the identified CEP128 missense variant (rs114668642). Population data indicate that MAF is 0.0234 and 0.0113 in the Japanese and Korean populations

(see https://www.ncbi.nlm.nih.gov/snp/rs114668642?horizontal_tab=true#frequency_tab; 3KJPN and KRGDB studies). The authors statement (page 7, lines 143-144) and Suppl. Table 1 apparently do not take into account such information and do not reflect the real picture. Text and Table should be revised taking in to account all the available population data;

A1: We thank the reviewer for kind reminder of the population genetic data from 3KJPN and KRGDB studies. These data have been added and further discussed in the revised manuscript. The population genetic data from the gnomAD, ExAC, and 1000 Genomes Project have been frequently used by geneticists to estimate variants' allele frequencies across human populations; therefore, we cited these data while we believe that the data from specific populations (as the reviewer kindly mentioned) are also very informative.

Q2: *The authors do not provide any information on the data output of the genomic analysis. This is a key aspect, particularly considering the consanguinity of the family. How many rare/private clinically relevant variants were annotated? How many rare functionally relevant homozygous variants? Did the authors exclude the possibility of other contributing cis/trans-acting events? Did the authors exclude absence of coverage (i.e. possible homozygous deletion) in coding exons of genes implicated in male infertility? (CNV?) Did the authors perform a SNP array analysis to exclude relevant LoH regions? (SNP array analysis).*

A2: By WES analysis, we identified 179 rare clinically annotated variants in the proband, and 174 in his sibling, while none of them have been associated with male infertility.

Moreover, 13 rare functionally relevant homozygous variants were found in the proband and his sibling respectively. Among them, five homozygous variants are shared by the proband and the affected sibling. These results have been shown in the supplementary dataset 1 in the revised manuscript. We then analyzed the possible functions of the genes affected by these brothers-shared homozygous

variants. Importantly, a homozygous missense variant in *CEP128* (c.665G>A [p.R222Q]) attracted our attention.

CEP128 is a centrosomal protein, and is primarily expressed in the testis. Centrosomal proteins are necessary components of the centrosome, which plays an important role in spermatogenesis. Additionally, *CEP128* is involved in the ciliogenesis. Therefore, we chose *CEP128* as the candidate gene for further functional studies. Importantly, the *in vitro* experiments further confirmed the deleteriousness of this mutation, and our mouse models carrying this *Cep128* knock-in mutation showed the reductions in sperm count and motility. These data thus indicate that this *CEP128*^{R222Q} mutation could be involved in male reproduction. Therefore, we did not perform other techniques for further analysis.

Generally, for the nonsyndromic infertile patients, we perform WES on them firstly. CNV analysis might be adopted under the circumstances that: (1) no candidate genes are detected by WES, (2) or strong evidences are supporting some genes related to the relevant phenotypes, while only the heterozygous variants were detected in them. Also, we cannot completely exclude the possibility of other contributing cis/trans-acting events, which are technically challenging due to the current limitation in annotation of non-coding variants. We fully agree with the reviewer's concern that consanguineous LoH regions potentially lead to homozygous recessive pathogenic variation. We pay close attention to recessive homozygous variation during the process of WES data analysis, which contributed to the discovery of candidate gene to a certain extent. Accordingly, more description of our analyses conducted in human subjects have been added into the revised manuscript.

Q3: Table 3 and main text (page 17 line 369 to page 18 line 376): it is not clear the genotype of the reported cases. Are they heterozygous or homozygous for the CEP128 changes? This is a relevant aspect;

A3: We stated "four heterozygous CEP128 variants were observed in five oligoasthenoteratozoospermia patients" in the previous manuscript (in line 372).

According to the reviewer's suggestion, we have supplemented the genotype of the four variants in Table 3 in the revised manuscript.

Q4: (4) The authors do not report the phenotype of the heterozygous KI mice. Presence/absence of relevant signs/features related to spermatogenesis/fertility should be provided. This is an important issue since the authors provide in vitro data apparently suggesting a "dominant" role of the R222Q CEP128 mutant. On the other hand, the homozygous condition in patients would suggest LoF, which is also in line with the "documented" fertility of the father.

A4: According to the reviewer's suggestion, we have added the phenotypic data of the heterozygous KI mice in Figure S8, Table S3 and Supplementary Movie 4 in the revised manuscript. The heterozygous KI mice are fertile. Papanicolaou staining and SEM showed the normal sperm morphology; sperm ultrastructure is also regular; and computer-assisted sperm analysis represented the normality in sperm count and sperm motility.

Other remarks:

Q5: Title: as previously noted by reviewer 1, the title does not reflect the major findings of this work. Based on the in vivo data, the authors may consider to revise the title to emphasize that "Disrupted/altered (or Loss of) CEP128 function impairs...";

A5: We thank the reviewer for expert suggestions. We have changed the title into "Disrupted CEP128 function impairs spermatogenesis".

Q6: Abstract (line 49): the identified variant is an annotated variant (rs114668642) (i.e., is not "novel"). The authors should revised the text accordingly.

A6: According to the reviewer's suggestion, we have revised this inappropriate description in the full text.

Q7: Abstract (line 52) and throughout the main text: the authors should specify that the KI mice are homozygous for the missense change;

A7: We have added “homozygous” to specify the KI mice.

Q8: From the pedigree reported in figure 2A, it is not clear the level of consanguinity. The authors should provide a complete pedigree of the family.

A8: We have modified the pedigree reported in Figure 2A in the revised manuscript.

Q9: Results/Tables: the authors should add the SNP IDs for the identified CEP128 variants.

A9: According to the reviewer’ suggestion, we have provided SNP IDs for the identified CEP128 variants in the Results and Tables in the revised manuscript.

If there are any additional questions/comments, please do not hesitate to let us know. We are willing to address them. Thank you very much!

Sincerely,

Ying Shen, Ph.D.

Professor, Department of Obstetrics/Gynecology, Key Laboratory of Obstetric, Gynecologic and Pediatric Diseases and Birth Defects of Ministry of Education

West China Second University Hospital, Sichuan University

P. R. China

REVIEWER COMMENTS

Reviewer #3 (Remarks to the Author):

The authors provided the missing controls for Fig S7 and satisfactorily answered all my questions.

I fully recommend the publication of this manuscript in its current form.

Reviewer #4 (Remarks to the Author):

This revised manuscript is an improved version of the previously submitted manuscript. However, some major concerns still stand and should be seriously considered by the authors.

(1) The authors do not discuss at all the relatively high frequency of the CEP128 R222Q variant reported in East Asian populations (MAF = 0.0234 and 0.0113 in Japan and Korea, respectively). Based on the size of those populations, you would expect >33,000 and 3,000 infertile males homozygous for that variant. This reviewer remains perplexed and confused about this picture. The authors should discuss on this specific criticism linked to the currently available population frequency data;

(2) Based on the consanguineous structure of this family, SNP array analysis in the two affected sibs should be performed to exclude putative homozygous structural rearrangements responsible/contributing to the trait. Moreover, it is not clear whether authors exclude absence of coverage (i.e. possible homozygous deletion) in clinically relevant genes?

(3) Based on the suggested recessive basis of this condition, the significance and relevance of the authors' finding related to the identification of 4 heterozygous CEP128 variants in 5 subjects from a cohort of 473 infertile individuals is relatively weak.

Reviewer #3 (Remarks to the Author):

The authors provided the missing controls for Fig S7 and satisfactorily answered all my questions. I fully recommend the publication of this manuscript in its current form.

A: We thank the reviewer for helping approve our work!

Reviewer #4 (Remarks to the Author):

This revised manuscript is an improved version of the previously submitted manuscript. However, some major concerns still stand and should be seriously considered by the authors.

A: We thank the reviewer for kind helps to improve our work. We have made significant revisions to our manuscript according to the expert comments. The detailed responses and changes are provided as follows.

(1) The authors do not discuss at all the relatively high frequency of the CEP128 R222Q variant reported in East Asian populations (MAF = 0.0234 and 0.0113 in Japan and Korea, respectively). Based on the size of those populations, you would expect >33,000 and 3,000 infertile males homozygous for that variant. This reviewer remains perplexed and confused about this picture. The authors should discuss on this specific criticism linked to the currently available population frequency data;

A: We thank the reviewer for kind reminder. The highest allele frequencies of the CEP128 R222Q variant was reported in the Japanese (0.5% to 2.34% according to the 1000 Genomes Project and 8.3KJPN project) and Korean populations (1.13% according to the KRGDB project), while this variant is rare or absent in the remaining populations of the world. For example, CEP128 R222Q was not detected in our 1000 normal Chinese control males from Sichuan, China. Moreover, the general population database (1000 Genomes Project) showed the frequency of this variant in Southern Han Chinese (CHS) is 0.0048, and it is absent in Han Chinese in Beijing (CHB) and African, European, and American populations. To explain the exceptional

distributions of the *CEP128* R222Q variant in the Japanese and Korean populations, we have the two following points that have been discussed in the revision.

(1) Male infertility is a common disease with a high incidence of approximately 7% in human populations. Therefore, it may be reasonable to observe the allele frequency of approximately 1% for a recessive pathogenic allele of male infertility (the frequency of the homozygote is approximately 0.01% in populations).

(2) A confirmed pathogenic variant can lead to the phenotypic variance of between populations. For example, the *gr/gr* deletion on the human Y chromosome is a significant risk factor for spermatogenic failure and this deletion is almost absent in the Dutch and some other Caucasian populations (Reppings et al. Nat Genet 2003). However, the prevalence of the *gr/gr* deletion varies significantly across different populations; and the highest frequency of the *gr/gr* deletion is in Japan (33.7%). These higher allele frequencies of disease-associated variants than expected in some specific populations may reflect the complex genetic mechanisms underlying the genotype-phenotype correlation, such as the impacts of genetic background and phenotype-modifier genes.

Certainly, we are sure that identifying *CEP128* R222Q mutation in more cases with male infertility and detecting other pathogenic/functional *CEP128* variants in other unrelated infertile patients will further support the causative role of *CEP128* deficiency in human spermatogenesis. Considering the helpful comments of the reviewer, we have toned down our descriptions of the *CEP128* R222Q variant and its potential roles in human spermatogenesis. Also, more discussions have been added into the revised manuscript accordingly. We believe that supplementing the discussion of specific populations (as the reviewer kindly mentioned) would help improving our manuscript quality.

(2) Based on the consanguineous structure of this family, SNP array analysis in the two affected sibs should be performed to exclude putative homozygous structural rearrangements responsible/contributing to the trait. Moreover, it is not clear

whether authors exclude absence of coverage (i.e. possible homozygous deletion) in clinically relevant genes?

A: We fully agree with the reviewer's concern about the potential contribution of homozygous deletion to the male infertility in this study. Here we paid close attentions to the absences of coverage that could be caused by homozygous deletion variations during the process of WES data analysis. No homozygous deletions of spermatogenesis related genes have been identified in this study. As suggested by the editor, more discussions and revisions have been made to the manuscript.

(3) Based on the suggested recessive basis of this condition, the significance and relevance of the authors' finding related to the identification of 4 heterozygous CEP128 variants in 5 subjects from a cohort of 473 infertile individuals is relatively weak.

A: Thanks for the reviewer's expert suggestion! In this study, our association analysis based on 473 infertile males and 1000 controls suggested the increased risk of male infertility in the heterozygous carriers of *CEP128* variants. However, functional studies are needed to investigate the potential pathogenic roles of these heterozygous *CEP128* variants. Also, our *Cep128* KO and KI mice showed the autosomal recessive inheritance of male infertility. Based on the current experimental evidence, we cannot readily confirm the roles of these heterozygous *CEP128* variants in male infertility. Therefore, we agree with the reviewer, we have removed this part from the Results section of the revised manuscript accordingly. Certainly, the risk of the heterozygous mutations of *CEP128* in male infertility needs to be further investigated.

If there are any additional questions/comments, please do not hesitate to let us know. We are willing to address them. Thank you very much!

Sincerely,

Ying Shen, Ph.D.

Professor, Department of Obstetrics/Gynecology,

Key Laboratory of Obstetric, Gynecologic and Pediatric Diseases and Birth Defects of

Ministry of Education

West China Second University Hospital, Sichuan University

P. R. China